# Revealing CO₂ dissociation pathways at vicinal copper (997) interfaces

Jeongjin Kim [1,6], Youngseok Yu[2,3], Tae Won Go[1], Jean-Jacques Gallet[4,5], Fabrice Bournel[4,5], Bongjin Simon Mun [2,3] ✉ & Jeong Young Park [1] ✉

Size- and shape-tailored copper (Cu) nanocrystals can offer vicinal planes for facile carbon dioxide ($CO_2$) activation. Despite extensive reactivity benchmarks, a correlation between $CO_2$ conversion and morphology structure has not yet been established at vicinal Cu interfaces. Herein, ambient pressure scanning tunneling microscopy reveals step-broken Cu nanocluster evolutions on the Cu(997) surface under 1 mbar $CO_2$(g). The $CO_2$ dissociation reaction produces carbon monoxide (CO) adsorbate and atomic oxygen (O) at Cu step-edges, inducing complicated restructuring of the Cu atoms to compensate for increased surface chemical potential energy at ambient pressure. The CO molecules bound at under-coordinated Cu atoms contribute to the reversible Cu clustering with the pressure gap effect, whereas the dissociated oxygen leads to irreversible Cu faceting geometries. Synchrotron-based ambient pressure X-ray photoelectron spectroscopy identifies the chemical binding energy changes in CO-Cu complexes, which proves the characterized real-space evidence for the step-broken Cu nanoclusters under CO(g) environments. Our in situ surface observations provide a more realistic insight into Cu nanocatalyst designs for efficient $CO_2$ conversion to renewable energy sources during $C_1$ chemical reactions.

Controlling the size and shape of the copper (Cu) nanocatalyst is a primary strategy for enhancing the reactivity and selectivity of carbon dioxide ($CO_2$) reduction reactions[1,2]. Computational predictions explain that the modulating micro-kinetics of tailored Cu morphologies play a critical role in highly improved $CO_2$ conversion toward $C_1$ or $C_2$ products[3,4]. However, revealing the catalytic reaction pathways of $CO_2$ and its intermediate is still challenging at the $CO_2$-Cu interface. The $CO_2$ dissociation to carbon monoxide (CO) and atomic oxygen (O) [$CO_2(g) \rightarrow CO* + O*$, asterisk: an adsorbate] begins with an endothermic step at Cu catalyst surfaces, owing to the highly positive enthalpy ($\Delta H = +293.0$ kJ/mol) thermodynamic aspect[5]. The $CO_2$ molecule has linear geometry in its electronic ground state, which has a limited chance of dissociative adsorption when the $CO_2$ molecule collides with close-packed metal atoms of terrace surface sites in low $CO_2$ pressure conditions[6]. Although the probed vibrational modes of $CO_2$ suggest a mechanistic understanding of catalytic $CO_2$ conversion at local step-sites of Cu(111)[7], our current knowledge about the $CO_2$ reduction process is still vague on the Cu nanocatalysts.

Meanwhile, synchrotron-based ambient pressure X-ray photoelectron spectroscopy (AP-XPS) indicated that the stepped Cu surface of Cu(100)[8] and Cu(997)[9] could allow the dissociation of $CO_2$ at step sites under 0.4 and 0.8 mbar $CO_2$(g) conditions, despite the low sticking probability of $CO_2$. According to the Gibbs−Helmholtz equation [$\Delta \mu = T\Delta S - kT\ln(p/p^0)$], the pressure gap beyond ten orders of magnitude may increase surface potential energy above 0.3 eV. Salmeron and colleagues[10,11] highlighted the fundamental principle of

¹Department of Chemistry, Korea Advanced Institute of Science and Technology (KAIST), Daejeon 34141, Republic of Korea. ²Department of Physics and Photon Science, School of Physics and Chemistry, Gwangju Institute of Science and Technology (GIST), Gwangju 61005, Republic of Korea. ³Center for Advanced X-ray Science, GIST, Gwangju 61005, Republic of Korea. ⁴Laboratoire de Chimie Physique-Matière et Rayonnement, CNRS, Sorbonne Université, Paris 75005, France. ⁵Synchrotron SOLEIL, Saint-Aubin, Gif sur Yvette 91192, France. ⁶Present address: Chemistry Division, Brookhaven National Laboratory, Upton, NY 11973, US. ✉e-mail: bsmun@gist.ac.kr; jeongypark@kaist.ac.kr

facilitating $CO_2$ dissociation over well-defined Cu surfaces and the subsequent restructuring of metallic Cu atoms at ambient pressure. It implies that the dissociated oxygen and CO from $CO_2$ would have also affected Cu morphologies under "working" conditions, unlike the conventional ultra-high vacuum (UHV) approach[12,13].

Here, we report the step-broken clustering of Cu atoms on the Cu(997) surface probed with ambient pressure scanning tunneling microscopy (AP-STM) under $CO_2(g)$ conditions. The $CO_2$ dissociation at vicinal Cu surfaces initiates the Cu morphology alterations under 1 mbar $CO_2(g)$. Time-lapse AP-STM observations show consistent results: sequentially restructured Cu atoms at step-edge sites and disordered Cu clustering phenomena. Our control experiments under $CO(g)$ or $O_2(g)$ conditions confirm that the observed step-broken Cu nanoclusters by repulsive CO–CO interactions show the reversible geometric transition, depending on the gas pressure conditions; once oxidized, the Cu(997) surface cannot be reverted to the original vicinal structure in UHV. Synchrotron-based AP-XPS measurements also indicate that the Cu(997) surface could have more enhanced CO adsorption coverage than that of Cu(111), which is caused by CO-induced restructuring of Cu atoms at step-edge sites under 0.2 mbar $CO(g)$. Our combined microscopic and spectroscopic evidence for adsorbate-driven morphology alterations on the Cu(997) surface emphasizes the critical role of vicinal structures on Cu nanocatalysts toward highly improved $CO_2$ reduction reactivity.

## Results

### Observations of Cu(997) morphologies at gas-solid interfaces

Schematic model illustrations of the Cu(997) surface are displayed in Fig. 1a and b. The indicated {997} facet has a 6.45° tilt against the (111) plane; the ideal step-interval and nearest-neighborhood distance correspond to 1.84 and 0.26 nm, respectively[14]. Figure 1c shows an as-

cleaned Cu(997) surface morphology after Ar⁺ ion-bombardment sputtering and annealing cycles in a UHV chamber. The measured step-interval is $1.8 \pm 0.5$ nm in the topographic profile analysis (Supplementary Fig. 1), which is close to the ideal {997} facet along the direction of [1 –1 2] on the STM image. The frizzy kinks at the Cu step-edge were occasionally observed during the STM scanning at 300 K (See Supplementary Text); the negligible feature did not exert itself on the topographic image.

We then introduced $CO_2$ gas into the STM reaction cell, where the gas pressure was constantly maintained at 1 mbar during AP-STM observations. Figure 1d exhibits an AP-STM image of the Cu(997) surface under 1 mbar $CO_2(g)$ (24 min lapse after introducing $CO_2$). The introduced $CO_2$ molecules cause the beginning of morphologic alterations from the step-edge sites of the Cu(997) surface at 300 K. We observed randomly reconstructed saw-like structures and broken steps simultaneously on the topographic images, as shown in Fig. 1e (77 min lapse after introducing $CO_2$). Time-lapse morphology observations also showed appearing-disappearing pictures of step-broken Cu nanoclusters along the direction of [1 –1 0] on the fixed scanning area as a time sequence (Supplementary Fig. 2). After pumping down the reaction cell in UHV (Fig. 1f), the probed Cu nanoclusters were no longer detected on the Cu(997) surface, except for the disordered and slightly widened step-edge sites.

In principle, linear $CO_2$ molecules collide with the metallic Cu surface via ineffective molecular collision trajectories. That can be a disadvantage of the C − O bond scission of $CO_2$ by charge transfers at the gas-solid interface. Nevertheless, there is a chance of bent-geometry transition caused by the increased surface potential energy at ambient pressure[8,15]. As a result, the bent-$CO_2$ likely prefers dissociative adsorption at step-edge sites of vicinal Cu surfaces, so that the dissociation progress of $CO_2$ molecules could create modified Cu

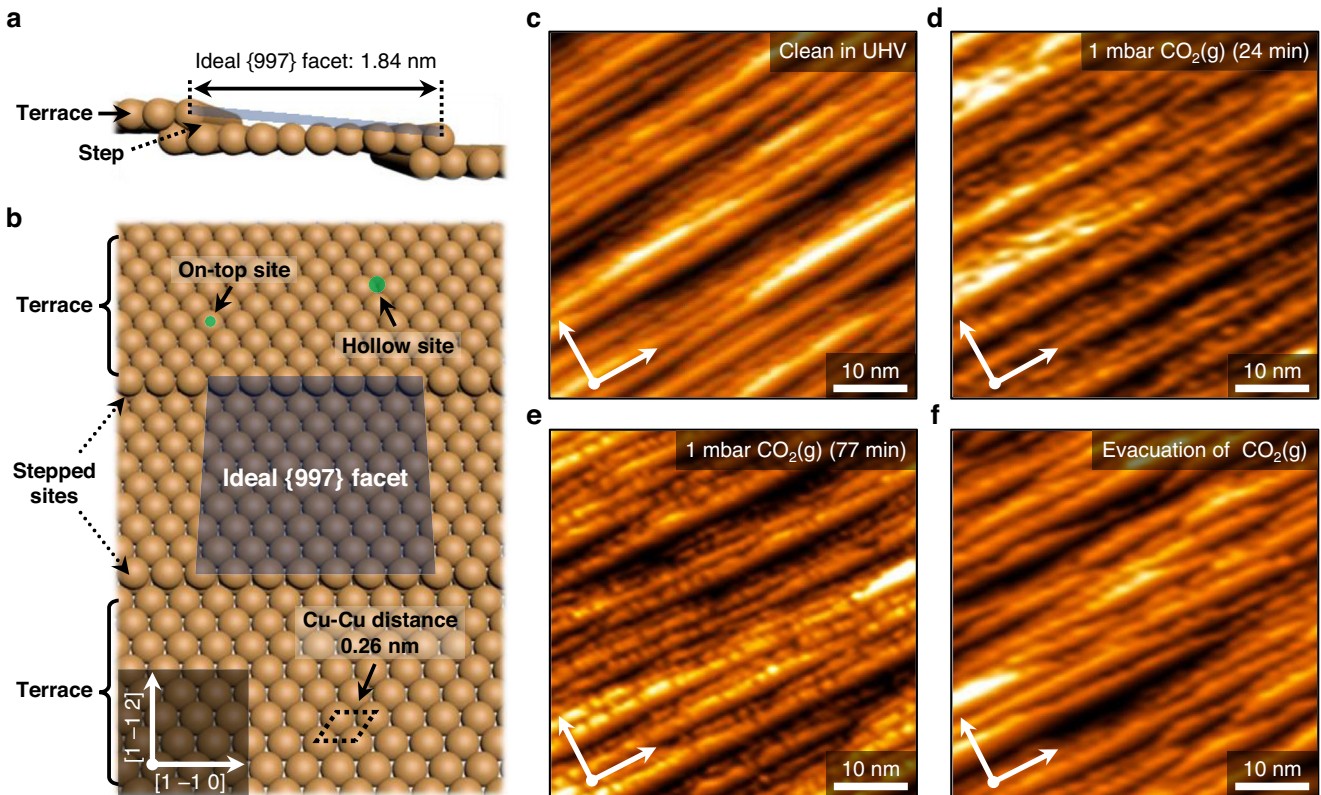

**Fig. 1 | The ideal Cu(997) surface structure and observed morphology alterations under 1 mbar $CO_2(g)$ (T = 300 K). (a, b)** Schematic illustrations of the ideal Cu(997) surface morphology. A dark orange ball represents a Cu atom [side view in (**a**) and top view in (**b**)]. (**c**–**f**) AP-STM images taken at $1 \times 10^{-10}$ mbar (**c**) [$V_s = 1.25$ V, $I_t = 200$ pA], 1 mbar $CO_2(g)$; t = $t_0$ + 24 min (**d**) [$V_s = 1.16$ V, $I_t = 180$ pA], 1 mbar $CO_2(g)$; t = $t_0$ + 77 min [$V_s = 1.24$ V, $I_t = 150$ pA], and after evacuation of $CO_2(g)$ (**f**) [$V_s = 1.28$ V, $I_t = 160$ pA] conditions.

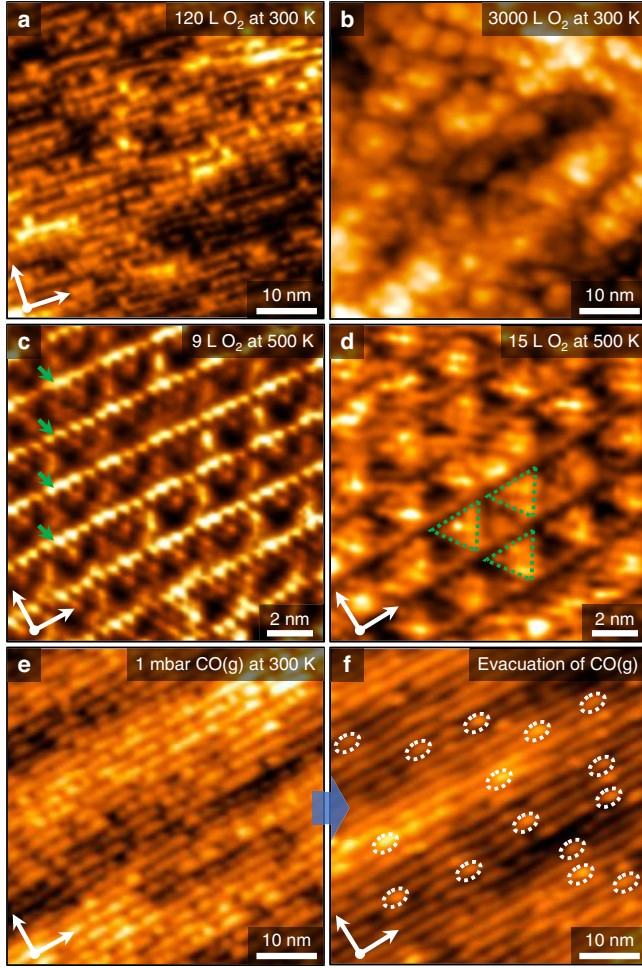

**Fig. 2 | AP-STM observations on the Cu(997) surface under O₂ or CO gas conditions. a–d** The oxidized Cu(997) surface structures after O₂(g) exposures in UHV. As-cleaned Cu(997) substrate was exposed to O₂(g) in the amount of 120 L (**a**) [$V_s$ = 1.25 V, $I_t$ = 200 pA] and 3000 L (**b**) [$V_s$ = 2.22 V, $I_t$ = 110 pA] at 300 K, respectively, before recording the topographic images in UHV. The observed triangular-shaped CuO$_x$ nanoclusters on the stepped Cu structure by oxidation pretreatments of 9 L (**c**) [$V_s$ = 1.58 V, $I_t$ = 120 pA] and 15 L (**d**) [$V_s$ = 1.25 V, $I_t$ = 200 pA] O₂ exposures at 500 K, respectively. Each green arrow on the STM image represents a row of dissociatively adsorbed oxygen at the step-edge site. The representative dotted green triangles on the STM image indicate the evolved CuO$_x$ clusters after O₂ gas exposures at 500 K. Both topographic images were recorded at 300 K. **e, f** AP-STM images of the reversible geometric transition by the pressure gap effect. The step-broken Cu nanoclusters form randomly along the direction of [1 –1 0] on the Cu(997) surface under 1 mbar CO(g) at 300 K (**e**) [$V_s$ = 1.10 V, $I_t$ = 200 pA], but almost all the Cu nanoclusters disappear after the evacuation of CO(g) in UHV (**f**) [$V_s$ = 0.94 V, $I_t$ = 130 pA]. Each dotted circle represents a remaining bump consisting of Cu atoms on the vicinal Cu surface.

morphologies, depending on the pressure level of the equilibrium gas[16]. The dissociated oxygen from CO₂ may be an effective source for the oxidizing process at the atomic level, which would have contributed to the site-specific disordering at step-edge sites caused by atomic O chemisorption[17].

Figure 2a–d show the oxidized vicinal Cu surface structures after variable amounts of O₂(g) exposure in UHV at 300 or 500 K. Even though the activation energy barriers of O₂ at the step and terrace geometries are similar, the dissociative adsorption pathway of O₂ precursors prefers Cu step-edge sites, because of the substantially different stabilization process of the transition state on the stepped Cu surface[18]. As displayed in Fig. 2a, we found reconstructed surface

morphologies and clustering of Cu atoms at the step-edge sites after O₂(g) leaked 120 Langmuir (1 L = 1.33 × 10⁻⁶ mbar · sec) into the UHV chamber. The topographic image demonstrates that the dissociated atomic O at the Cu step-edge could generate facetted saw-like CuO$_x$ structures on the vicinal Cu surface, which agrees with previous STM studies on Cu(111) and curved Cu crystals at 300 K[19,20]. We performed further oxidation of the Cu(997) surface at 300 K after O₂(g) exposures of 3000 L. The recorded STM image exhibits the formation of agglomerated Cu oxide clusters in Fig. 2b. The oxygen-driven surface reconstruction may initiate the growth of Cu₂O from the stepped Cu edge sites until they form oxide-like morphologies with a saturated oxygen coverage at equilibrium[21]. Thus, the high step density of the Cu(997) surface could enhance the initial oxidation rate of surface reconstruction progress, due to the facile nucleation process of the oxide precursors at step-edge sites.

The different kinetics of Cu₂O formations at 500 K explains the faceting stage on the vicinal Cu surface. In Fig. 2c, the green arrows indicate the adsorbed atomic O at the step-edge site by dissociating O₂ precursors. Moreover, Fig. 2d displays that the triangular-shaped CuO$_x$ nanoclusters are placed on the Cu steps without crumbling the stepped boundaries after O₂ exposures of 9 and 15 L. Such an initial oxidation process of the Cu oxide formation at the Cu step-edge site, which is crucial for analyzing atomic O adsorbate interactions with vicinal Cu structures. According to the combined study of theoretical calculations and AP-XPS experiments, the atomic O adsorbed on the Cu(100) surface could lead to the core-level shifts of binding energy, owing to the strong correlation between Cu and atomic O, even at submonolayer coverages[22]. In other words, even small amounts of atomic O should be incorporated with the Cu morphologies when CO₂ dissociation happens on the vicinal Cu surface. It is possible to discern the same rationale in our AP-STM observations as we probe the roughened Cu step-edge sites on the Cu(997) surface after evacuating 1 mbar CO₂(g) from the reaction cell (Fig. 1f). The dissociated atomic O from CO₂ at ambient pressure could have oxidized the stepped Cu geometries. Even though the geometric alteration process in Fig. 1f is more limited than the oxidized morphology at 300 K (Fig. 2a and b), the dissociative adsorption of oxygen occurs following the reaction probability of CO₂ (<-10⁻⁹/collision of CO₂ molecule)[6] at the slow kinetic rate during catalytic CO₂ dissociation.

The atomic O-covered Cu morphologies could be correlated with the irreversible reconstruction process at high oxygen coverages. Ultimately, these observed features are different atomic-scale events compared to the reversible Cu clustering at step-edge sites (Fig. 1e and f). The unexpected reversible Cu surface restructuring shows up more clearly during AP-STM observations under 1 mbar CO(g) conditions, as shown in Fig. 2e. The repulsive interactions between CO adsorbates may reorganize Cu steps to form Cu nanoclusters[23]. About 2 nm-sized Cu nanoclusters were observed in our real-space characterizations, and those recorded bumps on the topographic image did not entirely sustain their structures across the Cu(997) surface (Supplementary Fig. 3). Eventually, they disappeared immediately after CO(g) evacuation from the reaction cell, as shown in Fig. 2f. Although we found several remaining bumps in the same scanning area, the step-width changes were less than those under CO₂(g) evacuation conditions (Supplementary Fig. 4).

Our AP-STM observation results can be directly compared with previous literature regarding step-broken platinum (Pt) nanoclusters formation on the Pt(557) surface under CO(g) environments[24,25]. Of course, the chemical binding strength of CO-Cu is much weaker than that of CO-Pt, on account of insufficient π-electron back-donation from the metallic Cu to the CO adsorbate[26]. While the observed clustering trend of metallic atoms is analogous between Cu and Pt systems, the fundamentally correlated mechanisms during surface restructuring at elevated CO pressures must be different. Eren et al. claim that the CO-induced binding flexibility among metallic Cu atoms originates

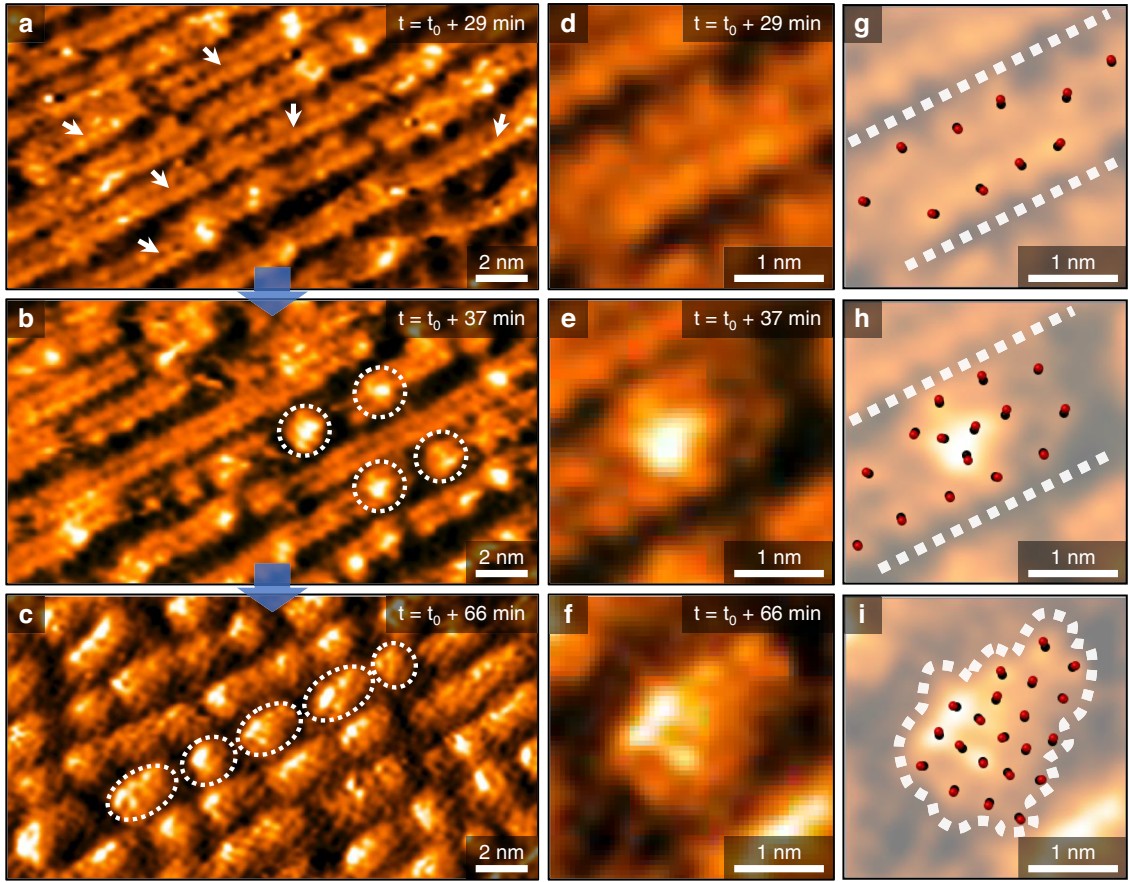

**Fig. 3 | Time-lapse morphology observations on the Cu(997) surface under 1 mbar CO gas conditions (T = 300 K). a–c** AP-STM images taken at 29 (**a**), 37 (**b**), and 66 min (**c**) after filling the reaction cell with CO gas [$V_s$ = 1.10 V, $I_t$ = 200 pA]. White arrows and dotted circles correspond to CO adsorbates at step-edge sites and evolving Cu nanoclusters on the topographic images. **d–f** Magnified local morphologies from the AP-STM images taken at $t_0$ + 29 (**d**), $t_0$ + 37 (**e**), and

$t_0$ + 66 min (**f**). **g–i** Corresponding CO-bound maps on the AP-STM images taken at $t_0$ + 29 (**g**), $t_0$ + 37 (**h**), and $t_0$ + 66 min (**i**), respectively. Dotted lines and a closed curve correspond to under-coordinated Cu sites and an evolved Cu nanocluster. Combined black (carbon atom) and red (oxygen atom) balls on the image illustrate CO adsorbates.

from a significant gap in the cohesive energy between Cu and Pt (Cu: 3.49 eV/atom, Pt: 5.84 eV/atom)[27,28]. The CO adsorbates may lift Cu atoms above half of the monatomic height, even at a relatively low CO coverage on the Cu(111), compared to the case of CO/Pt(111). So, the formation of CO-Cu species would facilitate further growth of $Cu_x(CO)_y$ clusters with a specific mesoscopic mismatch in the surface region[29].

Nevertheless, the dissociated CO from $CO_2$ would be a minor species compared to atomic O, because of its low adsorption energy on Cu terraces. For example, previous AP-XPS studies on $CO_2$/Cu(100) mainly highlighted the atomic O blocking on Cu catalysts rather than CO chemisorption, because the covered oxygen species would hinder the yield of $CO_2$ conversion toward methanol synthesis[8,16]. Another possible pathway of carbonate ($CO_3$) formation from atomic O and $CO_2$ cannot be overlooked in terms of thermodynamics. Koitaya et al. discussed the detected atomic O species and their role in the reaction mechanism of $CO_2$ dissociation on the Cu(997) surface under 0.8 mbar $CO_2$(g) conditions[9]. They claimed that the dissociated atomic O could react with a free molecule of $CO_2$ to form a negatively charged $CO_3^{\delta-}$ adsorbate [$O^* + CO_2(g) \rightarrow CO_3^*$] at 340 K. Overall, their explanation can justify the sophisticated reaction mechanism of the $CO_2$ conversion over Cu catalysts, assuming low adsorption energy of CO on low-index Cu surfaces. However, CO adsorbates could also have stabilized at under-coordinated sites of the vicinal Cu surface due to the enhanced CO binding energy[23].

## Formation of step-broken Cu nanoclusters at 1 mbar CO(g)

Figure 3a–c demonstrate time-lapse AP-STM images of the reorganizing stepped morphologies on the Cu(997) surface under 1 mbar CO(g). White arrows in Fig. 3a (29 min lapse after introducing CO) indicate noticeable atomic-scale corrugations at step-edge sites. Their nearest-neighboring distance is close to 0.52 nm from line profile analysis (Supplementary Fig. 5), which is a double atomic length of Cu-Cu (0.26 nm), as shown in Fig. 1b. The CO adsorbates typically occupy on-top positions of metallic Cu atoms on the stepped Cu surface[30,31], developing the active progress of step-broken clustering from CO-Cu chemical binding at step-edge sites. We can see a few nanoclusters of about 2 nm (white-dotted circles) around the broken Cu facets in Fig. 3b (37 min lapse after introducing CO) during the AP-STM observations with a fixed scanning area. Lastly, the step-broken Cu nanoclusters appear widely on the Cu(997) surface, as shown in Fig. 3c (66 min lapse after introducing CO). We could characterize the lateral size of evolved Cu nanoclusters (2.1 ± 0.6 nm) from the AP-STM image and statistical analysis (Supplementary Fig. 6).

The CO-induced clustering correlates to the attractive and repulsive interactions between CO adsorbates within femto- ($10^{-15}$) or pico- ($10^{-12}$) seconds, depending upon Cu surface structures[32,33]. In addition, the CO chemisorption energy at under-coordinated Cu atoms would have increased to 0.35 eV[34]. Even though those instant molecular interactions happen in the local cross-sectional area of less than 1 Å$^2$, their consistent influences may have gradually reorganized

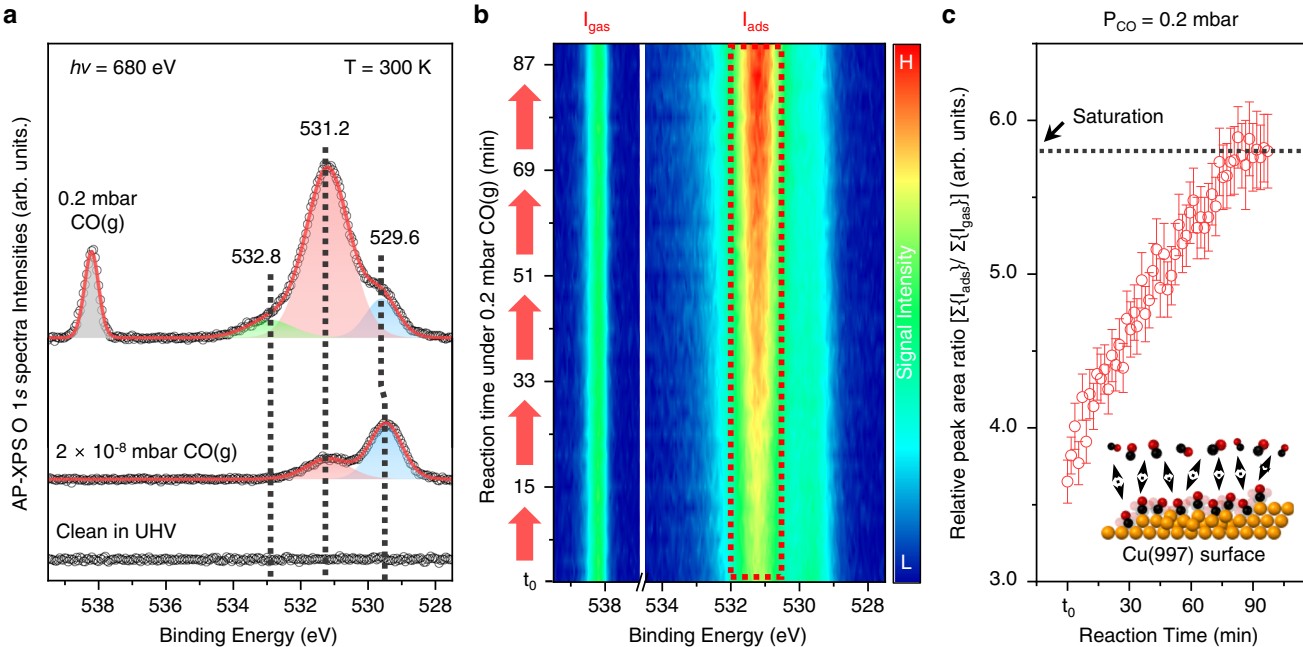

**Fig. 4 | Synchrotron-based AP-XPS measurements on the Cu(997) surface (T = 300 K). a** Collected O 1s core-level AP-XP spectra (hv = 680 eV) in UHV (bottom) and under CO gas conditions of $2 \times 10^{-8}$ (middle) and 0.2 mbar (top). **b** Contour map analysis of time-lapse measurements for O 1s core-level spectra after filling the analysis chamber with CO gas ($P_{CO} = 0.2$ mbar). **c** Estimated relative peak area ratio of $I_{ads}/I_{gas}$ plots from the time-lapse AP-XPS measurements.

Cu atoms at the vicinal surface structure to reduce surface free energy in a gaseous environment on a time scale of minutes[12,35].

Figure 3d–f show the magnified AP-STM images from the selected local surface areas in Fig. 3a–c, exhibiting the selected CO-induced Cu nanocluster formation. The bright protrusions on each enlarged topographic image may represent $Cu_x(CO)_y$ complexes on the Cu(997) surface. The proposed CO-Cu configurations could assist our understanding of the atomic-scale clustering of Cu atoms at 1 mbar CO(g), as visualized in Fig. 3g–i. We can only observe atomic-scale Cu bumps around step-edge sites in Fig. 3g, whereas the agglomerated Cu nanoclusters and discernible step-broken regions are found in Fig. 3h and i. Notably, an enclosed white-dotted curve in Fig. 3i indicates mostly dense $Cu_x(CO)_y$ complexes. The measured neighboring distances of bright protrusions are varied within 0.3 – 0.4 nm on the enlarged image. They would have had different CO adsorption energies of −0.34 through −0.84 eV on the stepped Cu surface, depending on the clustering size[23]. Such CO chemisorption should make more segmented surface free energy distributions in different coordination numbers of Cu atoms by molecular interactions on the surface[36], resulting in chemical binding energy changes on the Cu(997) under CO(g) conditions.

We note that the observed step-broken clustering is not involved in the gas contamination issue with nickel (Ni) carbonyl-containing compounds. Our AP-STM images demonstrate the reversible Cu(997) morphology alterations by repulsive CO–CO interactions. The observed clustering features under CO(g) conditions were not caused by well-known surface contaminations by step-oriented oxidation[20] nor accumulated $Ni(CO)_4$[37] compounds. The reversible clustering phenomenon at the Cu step-edge originates from the force gradient at equilibrium energy distance by repulsive CO interactions[33], which leads to the vertical displacement of Cu atoms depending upon variable CO adsorption energies at the step-edge sites. In fact, the observed Cu nanoclusters on the Cu(997) surface also have a discrepancy with the CO-driven clustering results on the Cu(111) surface. The size-dependent mesoscopic mismatch[29] is confined on the Cu(997) compared to the Cu(111) surface, which would allow relatively limited mass transport during the clustering of Cu atoms at step-edge

sites. In short, the lifted Cu nanoclusters along the vertical direction cannot be fully extracted from Cu steps, unlike the irreversibly reconstructed Cu(111) surface under 1 mbar CO(g) (Supplementary Fig. 7). Besides, we can exclude the influence of residual hydrogen ($H_2$) gas in the UHV chamber, because we could not observe any significant Cu atom restructuring process on the Cu(997) surface under 1 mbar $H_2$(g) at 300 K (Supplementary Fig. 8).

**Synchrotron-based AP-XPS analysis of CO-Cu species**

Furthermore, we carried out synchrotron-based AP-XPS measurements to investigate the specific property of CO adsorbates on the Cu(997) surface under CO gas environments at the TEMPO beamline of synchrotron SOLEIL facility in France. Our spectroscopic analysis setup could detect modifications of core-level states on the vicinal Cu surface during CO-induced morphologic changes at ambient pressures. We prepared a clean Cu(997) surface in UHV by the same crystal cleaning protocol in the AP-STM observation. The prepared clean sample was quickly transferred into an analysis chamber without vacuum loss.

Collected O 1s core-level AP-XPS signals at selected photon energy (hv = 680 eV) are plotted in Fig. 4a as a function of backfilling CO(g) pressure in the analysis chamber. The inelastic mean free path of the analyzed photoelectron was kept within three sublayers from the topmost layer of the Cu(997) substrate. A contaminant-free spectrum acquired in UHV sensitively changed after $2 \times 10^{-8}$ mbar CO(g) gas leaked into the analysis chamber. The resolved peaks of 529.5 and 531.2 eV in the O 1s core-level spectrum are associated with the results of CO chemisorption on the vicinal Cu surface. The probed peak intensity at 529.5 eV is higher than the adjacent peak at 531.2 eV by a factor of 2.7.

In contrast, we confirmed a clear rising trend of peak intensity at 531.2 eV and a peak shift of +0.1 eV from 529.5 eV during O 1s core-level measurements under the 0.2 mbar CO(g). The observed chemical species at 529.5 eV was not identified in O 1s core-level spectra of the Cu(111) surface in the same experimental environment (Supplementary Fig. 9). According to previous literature, the appeared peak or spectral shoulder at 529.5–529.6 eV would be assigned to atomic

oxygen[38] by dissociation of CO or $H_2O$[28,39]. The introduced CO molecules may have interacted with a limited number of defect sites at the early stage of the overall chemical reaction over the vicinal Cu facets. As predicted by quantum mechanical calculations, CO molecules prefer adsorption on under-coordinated Cu atoms over close-packed Cu atoms on terrace regions to compensate for thermodynamic stability[31]. So, the CO chemisorption at metallic step-edge sites has relatively electron-rich states, depending on the CO coverage in XPS analysis[40].

The total amount of atomic oxygen species at 529.5–529.6 eV should be small compared to the significant peak evolution at 531.2 eV, which is supported by the recorded survey spectrum, indicating a substantial difference in peak intensities between Cu 2p and O 1s (Supplementary Fig. 10). The specified region of the peak evolution at 531.0–531.2 eV might also be attributed to the adsorbed water/hydroxyl species on the Cu surface, in accordance with previous reports. In general, molecular water does not prefer sticking onto the Cu terraces, which leads to thermodynamically metastable adsorption states on the Cu surface[41]. The oxygen-pre-covered Cu surface can provide a much more favorable environment for molecular water adsorption at a higher relative humidity (5%; T = 295 K)[42], but our AP-XPS operating condition (P = 0.2 mbar CO; T = 300 K) is far from such a humid environment.

In fact, the characterized species at 531.2 eV could originate from the roughened Cu surface by forming CO-Cu species; that physicochemical phenomenon was also clearly reported elsewhere by AP-XPS measurements[28,43] under CO(g) conditions at room temperature. It was also possible to characterize a broadened spectral tail around 532.8 eV that originated from the CO chemisorption on kink and corner Cu atoms during the surface restructuring[24]. Thus, the identified spectral features at 531.2 and 532.8 eV can be assigned to the chemisorbed CO in Fig. 4a, and we also find coexisting CO-Cu species[39] at 285.6 and 287.7 eV in the C 1s core-level spectrum at that time (Supplementary Fig. 11). Based on interpretations of CO/Cu system in previous reports, it is rational that we assign the resolved peak at 531.2 eV as inner-sites of Cu step-edges on {997} facets, because the clustering phenomenon under ambient CO(g) environments could be related to the formation of CO-Cu species on the Cu surface.

The geometric transition can affect the electronic structure and step-step interaction on the vicinal surface[44], which leads to the changes in energetics of CO chemisorption between flat and defective sites[45]. Although the Cu surface has weaker CO-Cu bonding than that of CO-Pt, the under-coordinated Cu atoms have significantly reduced charge density compared to the flat Cu sites. So, the increased CO binding energy enhances the elastic step-step interactions[46] on the Cu(997) surface, due to the anisotropic Smoluchowski effect[47]. The difference in charge distributions induces correlated electron dynamics pathways between initial and final states on the various flat and under-coordinated Cu sites, possibly observed as frustrated CO chemisorption[48] in spectroscopic analysis. Those recorded pressure-dependent changes implicate that the CO-induced vicinal Cu structure had morphologic alterations by increasing CO coverage at ambient pressures.

Figure 4b exhibits more details of identified spectroscopic changes from collected O 1s core-level spectra caused by CO adsorbates on the vicinal Cu surface at 0.2 mbar CO(g). We successively plotted the time-lapse monitoring signals on the color mapping of signal intensity at binding energy versus reaction time after filling the analysis chamber with CO gas at 300 K. It is evident that a remarkable enhancement of peak intensity at 531.2 eV occurred on the contour map analysis, at the elapsed reaction time of 0.2 mbar CO(g). Selected O 1s core-level spectra, acquired at initial reaction time ($t_0$), $t_0 + 47$, and $t_0 + 92$ minutes, show the peak intensity variation at each recorded moment (Supplementary Fig. 12). In the meantime, we only identified subtle spectral alterations at 529.6 and 532.8 eV in the O 1s core-level

spectra—their signal intensities are nearly identical throughout the spectroscopic analysis between $t_0$ and $t_0 + 92$ minutes. The chemical state changes are analyzed from the increased CO coverage on the vicinal Cu surface.

We estimated the composition ratio of O/C from the integrated peak areas for assigned CO species in O 1s and C 1s core-level spectra at 0.2 mbar CO(g). After normalizations of the AP-XP spectra by peak intensities of CO(g), we compared the integrated peak areas of assigned CO-Cu species in O 1s (531.2 and 532.8 eV) and C 1s (285.6 and 287.7 eV) spectra. The relative portion of $sp^2$ (C = C) or $sp^3$ (C – C) carbon species in the C 1s core-level spectrum was not considered for this analysis to focus on the detected CO-Cu features in the fingerprint regions of –CO functional groups. In this quantitative analysis, we obtained a reliable value of 1.1 ± 0.1 for the atomic ratio of O/C, which confirms the peak evolutions by molecular interactions of CO adsorbates on the Cu(997) surface. Technically, we cannot thoroughly exclude the influence of the photoelectron diffraction effect and differences in core-level ionization cross-section between O 1s and C 1s spectra at different photon kinetic energies. Despite those experimental uncertainties, the empirical analysis method is useful for estimating the relative compositional ratio of O/C from the collected core-level spectra[9].

Further investigation in X-ray photoelectron spectra for Cu 2p core-level state and X-ray absorption spectra (XAS) on Cu $L_{2,3}$ absorption-edge indicates that the characterized metallic Cu surface ($Cu^0$) does not make a transition to layered oxides consisting of $Cu_2O$ ($Cu^+$) or CuO ($Cu^{2+}$)[49] with prolonged CO(g) exposure (Supplementary Fig. 13). We found a small peak at 946.4 eV in Cu 2p core-level spectrum under 0.2 mbar CO(g), but the detected feature disappeared after gas evacuation from the analysis chamber in UHV. The suspected shake-up feature is correlated to significant ligand–metal charge transfer at the gas-solid interface, rather than the existence of lattice oxygen[43]. Besides, the obtained XAS in UHV, 0.2 mbar CO(g), and after gas evacuation conditions do not match with referenced data of Cu oxides[50]. Even though we measured a minor change of signal intensities at the Cu $L_3$-edge under 0.2 mbar CO(g), the enhanced peak intensity was attenuated again, the same as the original species after gas evacuation.

Technically, the accepted value of probing depth by XAS is approximately 2 nm in Auger electron yield (AEY) mode[51,52]. In previous literature, we find a similar feature in the pronounced resonance peak due to the formation of a thin (~1 nm) $Cu_2O$ layer during XAS measurements under CO oxidation conditions [p{$O_2$} = 0 – 0.2 mbar; p{CO} = 0.4 mbar][53]. The spectroscopic feature of Cu(I) or Cu(II) oxides can be easily distinguished in regions of the Cu surface and sub-surface because of the thermodynamically favored penetration of adsorbed oxygen[54]. With the nature of Cu-O and the basis of the AEY-XAS principle, our XAS results collected before and after introducing 0.2 mbar CO(g) exhibit no significant evidence for the formation of Cu(I) or (II) oxide structures in the near-surface layers of the vicinal Cu crystal. However, the sensitive change of Cu $L_{2,3}$-edge is only detectable at 0.2 mbar CO(g). A kinetic Monte Carlo simulation study[55] provides theoretical insight regarding the unusual feature in the first resonance peak of Cu $L_3$-edge under CO(g) conditions. Because the CO-driven migration of surface Cu atoms at under-coordinated sites caused the creation of an unusual type of active site, the unique electron transfer property by CO adsorbates at 0.2 mbar CO(g) would have been involved in the transient changes in our XAS measurements.

Figure 4c displays the estimated relative peak area ratio of $I_{ads}/I_{gas}$ from our AP-XPS data during Cu(997) surface restructuring. The displayed plots clearly indicate the evolution of CO-Cu species at the vicinal Cu interface by a factor of ~5.8 at the saturation, whereas the Cu(111) surface only shows minor changes before and after the saturation of ligand coverages at 0.2 mbar CO(g) (Supplementary Fig. 14). It means that the repulsive CO – CO interactions may have

driven surface restructuring at vicinal Cu surfaces compared to Cu(111) under the same experimental conditions. Likewise, our AP-STM image shows no significant surface restructuring on the Cu(111) substrate under 0.1 mbar CO(g). We observed several faceting sites by CO adsorbates at step-edge sites, but those local geometric alterations did not propagate to such wide surface restructuring (Supplementary Fig. 15).

According to simulated thermodynamic values in the literature, the clustered CO-Cu complexes have low formation energy (0.39 eV). The Cu surface could break up into Cu nanoclusters by repulsive interactions between adsorbed CO molecules, reducing the surface free energy at ambient pressure[23]. The aggressive CO chemisorption environments weaken metallic binding among Cu atoms. The potential energy difference between $2 \times 10^{-8}$ and 0.2 mbar CO(g) is roughly 0.4 eV, which is enough to activate the surface reconstruction on the Cu(997) surface. The thermodynamically unstable vicinal surface structures can easily redistribute the compressed tensile stress of stepped sites by rearranging the atomic length of metallic coordination at a saturation CO coverage[56]. Those unusual properties of CO-Cu interactions may cause the observed unexpected surface restructuring phenomena on the vicinal Cu surface as a function of CO pressures. Consequently, the CO adsorbates would induce successive migrations of stepped Cu morphology, consisting of under-coordinated Cu atoms at the gas-solid interface, resulting in the dramatic growth of CO coverage caused by the Cu nanoclusters' evolution[11].

### Catalytic behaviors of dissociated O and CO adsorbates from $CO_2$

As shown in previous sections, the surrounding gaseous environmental conditions of $O_2$ or CO molecules affect the irreversible surface reconstruction or reversible restructuring phenomena on the vicinal Cu surface. In principle, the dissociative adsorption of atomic oxygen and CO adsorbates from $CO_2$ could be involved in the observed geometry alterations at the Cu step-edge sites (See Supplementary Text). Obtained AP-STM images and AP-XP spectra under the different gaseous conditions could provide consistent evidence for the complementary relations of adsorbed atomic oxygen and CO adsorbates on the Cu(997) surface, and they suggested atomic-scale insights regarding the unexpected surface restructuring under ambient pressure environments. Theoretically, the catalytic reaction pathway of $CO_2$ conversion is very complicated on Cu-based catalysts, and the dissociation yield of $CO_2$ could be influenced depending on surface geometries at the early stage of overall catalytic reactions[8,57,58]. The catalytic dissociation of $CO_2$ can simultaneously produce atomic oxygen and CO from the Cu step-edge sites. However, fundamental knowledge of adsorbate molecule interactions at the gas-solid interface has mainly been discussed in metallic or O-covered Cu surfaces with low-index facets in previous literature[16,59–61]. The extension of principles from only the low-index facets such as {111}, {110}, and {100} may have a restricted point of view to represent the formation of active sites on rational nanocatalyst designs[62–64] consisting of vicinal surface geometries and ligand interactions during $CO_2$ activation processes.

Our spectroscopic and microscopic analysis results reveal the critical role of adsorbate interactions with $CO_2$ dissociation at under-coordinated Cu sites, as shown in comparison results probed with AP-XPS and AP-STM (Supplementary Fig. 16). Obviously, we find different trends of the Cu clustering under $CO_2$(g) conditions compared to the observed Cu(997) surface exposed to $O_2$(g) or CO(g), as shown in Figs. 2 and 3; i.e., there is a mixed fashion of morphologies between oxidized Cu step-edges (Fig. 2a) and CO-driven Cu nanocluster formations (Fig. 3c) on vicinal Cu surfaces. According to the reported synchrotron-based AP-XPS measurements, the formation of $CO_3^*$ species was much more pronounced than atomic oxygen in O 1 s core-level AP-XP spectra under 0.8 mbar $CO_2$(g) at 340 K[9]. In the present AP-XPS study (Supplementary Fig. 16), we also detected both atomic

oxygen and $CO_3^*$ peaks caused by $CO_2$ dissociation at 300 K, but plotted photoelectron signals in the O 1s core-level spectra indicate an opposite trend in the characterization of adsorbate species compared to the previous literature. This could originate from the technical issue of characterizing $CO_3^*$ species, due to the different probing depths between the synchrotron-based photon source ($hv = 630$ eV) and monochromatic Al $K\alpha$ X-ray source ($hv = 1486.7$ eV), as well as other experimental conditions such as gas flow rate, sample temperature, lapsed-time, and background pressure in the analysis chamber.

Overall, the dissociated O* from $CO_2$ at Cu step-edge sites may cause irreversible surface roughening due to the formation of atomic-scale $CuO_x$ clusters. Consequently, the following chemical reaction between the adsorbed atomic O* and $CO_2$ molecule may evolve $CO_3$-related chemical species at the Cu step-edge sites, affecting local surface restructuring during the catalytic $CO_2$ activation at ambient pressures. These atomic-scale behaviors under $CO_2$(g) conditions differ from the reported CO-driven surface restructuring phenomena caused by repulsive CO–CO interactions[24,25,28], as the dissociated O* or $CO_3^*$ species also participate in the surface geometry alterations at the step-edge sites of vicinal Cu surfaces.

## Discussion

Our atomic-scale in situ characterizations, combining AP-STM and synchrotron-based AP-XPS observations, provide convincing experimental information for the evolution of Cu nanoclusters on the Cu(997) surface by single carbon molecules. Such analysis results suggest a resolution to the long-standing curiosity about vicinal Cu surfaces and the elusive behavior of Cu adatoms at the gas-solid interface. The surface-molecule interaction of $CO_2$ at narrow-stepped Cu morphologies can effectively facilitate the dissociation of $CO_2$, producing atomic oxygen and CO adsorbates on the vicinal Cu surface at ambient pressure. The dissociated species can competitively occupy under-coordinated sites on the vicinal Cu(997) surface, and these molecular behaviors accompany the complicated surface restructuring phenomena with the formation of kinks and Cu nanoclusters around step-edge sites. CO adsorbates are particularly able to propagate wide rearrangements of Cu atoms from the step-edge sites of the metallic Cu surface by repulsive CO–CO interactions at the gas-solid interface. Eventually, the clustering phenomenon happens widely across the Cu(997) surface, because the metallic Cu atoms could be lifted by CO adsorbates, but they are not fully ejected from the facet of vicinal Cu morphologies, due to the relatively weak binding strength between Cu and CO at 300 K. As a result, we find unusual reversible surface restructuring on the vicinal Cu surface as a function of adsorbed CO coverage. That is distinct from the oxygen-covered geometries, in that the oxidation process of the Cu surface is generally accomplished by irreversible surface reconstruction. Overall, the observed surface roughening trends caused by atomic oxygen and CO adsorbates may affect the result of successive catalytic reactions to produce critical intermediates, such as carbonate and formate, during the $CO_2$ reduction reaction. Indeed, those investigation results provide convincing evidence for the influence of surface morphology structures at the atomic level. Our findings indicate that controlling the Cu nanoclusters would establish an efficient $CO_2$ reduction reaction in working conditions. The bent-$CO_2$ dissociation over Cu nanocatalysts consisting of high-index facets may open a prospect for improving conversion yields of $CO_2$ to sustainable future energy sources in industrial processes.

## Methods
### Sample preparation
Commercially available Cu single crystals (99.999%) were purchased (disc-shaped diameter – 8 mm; thickness – 1 mm; cut orientation accuracy <0.1°; the roughness of surface <0.03 μm; one-side polished) from the metal single crystal growth manufacturer (Mateck GmbH,

Germany). A polished side of the crystal plane was further pretreated by the surface cleaning protocol in UHV. The Cu(111) surface was cleaned by cycles of Ar⁺ ion-bombardment sputtering ($P_{Ar} = 1 \times 10^{-5}$ mbar; Accelerated voltage = 800 eV) for 20 min and substrate annealing at 800 K for 5 min. We repeatedly performed this procedure until the Cu surface had no contaminants. The stepped Cu(997) single crystal was pretreated to remove carbon and oxygen impurities at step-edge sites by the same surface cleaning protocol. Then, we had an additional cycle of Ar⁺ ion-bombardment sputtering ($P_{Ar} = 3 \times 10^{-6}$ mbar; accelerated voltage = 600 eV) for 10 min and then annealing at 600 K for 5 min. Both sample pretreatment procedures were carried out equally in separated surface analysis systems for AP-STM observations (KAIST, the Republic of Korea) and synchrotron-based AP-XPS measurements (Synchrotron SOLEIL, France).

## AP-STM measurements

We obtained surface morphology images at 300 K using an inchworm-type STM scanner inside the reaction cell (Aarhus STM 150 NAP, SPECS GmbH). A sample stage enclosed by a small dome-shaped space (15 mL) could be isolated from the UHV chamber ($P_{base} = 1 \times 10^{-10}$ mbar) by two Viton O-rings seals and locking screws. The compact-sized reaction cell was suspended from the STM flange by using three springs to ensure vibrational damping control in the system. The high-purity CO gas (99.999%) was passed through a metal-carbonyl removal filter (PALL Corporation, the United States) for further gas purification. After then, the purified gas was slowly introduced into the STM reaction cell using a precision leakage valve. A chemically-etched tungsten tip was adopted for surface morphology observations. The topographic images were recorded by tunneling current measurements between the STM tip and positively biased sample in constant current mode. Feedback loop controllers accurately manipulated the STM tip during the direct observations. We had control of the STM tip movements within the limit of the sub-nanometer in three-dimensional space. Each recorded topographic image represents the empty state of the measured surface electronic structure. The corresponding tunneling condition of the STM image is denoted as $V_s$ (applied sample voltage) and $I_t$ (tunneling current). We had a bakeout procedure (T = 398 K) on the AP-STM chamber and on the home-built gas manifold for 24 hr to prevent residual gas contaminations just before surface morphology observations under the gas environments.

## Synchrotron-based AP-XPS and AP-XAS measurements

Synchrotron-based XPS experiments were carried out in the AP-XPS setup managed by the LCPMR group (Sorbonne Université), attached to the TEMPO beamline of the French national synchrotron facility (SOLEIL) in France. The synchrotron-based experimental setup consisted of a load-lock ($P_{base} = 5 \times 10^{-7}$ mbar), a distribution ($P_{base} = 1 \times 10^{-9}$ mbar), a preparation ($P_{base} = 3 \times 10^{-10}$ mbar), and an analysis ($P_{base} = 5 \times 10^{-10}$ mbar) chamber. As-cleaned Cu crystal was transferred between chambers without vacuum loss by transfer rods. Then, we loaded the sample onto a multi-axis manipulator in the analysis chamber. The polished side of the Cu crystal was positioned in a perpendicular direction to the cone-aperture ($\phi = 300$ μm) of a hemispherical electron analyzer (Phoibos 150 NAP, SPECS). An X-ray beam irradiated the Cu surface through a differentially-pumped windowless-beam entrance design at a fixed angle (a 54° tilt against the hemispherical electron analyzer). At the synchrotron radiation facility, the incident photon energy resolution ($h\nu/\Delta h\nu$) was ~5000. The optimized traveling distance of the photoelectron between the Cu surface and cone aperture was 0.5 – 1.0 mm during the AP-XPS measurements. The escaped photoelectrons were concentrated using an electrostatic lens system and differential pumping stages. The core-level AP-XP spectra were collected after filling the analysis chamber with high-purity CO gas (99.999%). X-ray absorption spectra (XAS) of the Cu $L_{2,3}$-edge were recorded between 925 and 965 eV at a photon energy grating step of

0.1 eV in Auger electron yield (AEY) mode. We strictly kept the signal sensitivity ratio between chemisorption CO species and gas-phase CO. The filled CO gas was monitored by a quadrupole mass spectrometer (QMS) in the second differential pumping stage. Each core-level state spectrum was calibrated against the Fermi-edge of Cu single crystal at selected photon energy. All core-level AP-XP spectra were fitted by a mixed Gaussian−Lorentzian function (70%: 30%) after the Shirley-type background subtraction using a CasaXPS package. The saturation coverage of CO(ads.) was estimated from the (1.4 × 1.4)–CO super-structure on the Cu(997) surface ($\theta_{CO} = 0.52$ molecules/one surface Cu atom)[65].

## Laboratory-based AP-XPS measurements

The home-built AP-XPS system with a commercially available X-ray source (a monochromatic Al $K\alpha$ anode; $h\nu = 1486.7$ eV) was utilized for investigations of Cu $2p$, O $1s$, and C $1s$ core-levels changes on the Cu(997) surface under 0.8 mbar $CO_2$(g) conditions. The GIST-ESCA group managed and conducted the laboratory-based AP-XPS setup (GIST, the Republic of Korea). The adjusted incident angle of the X-ray beam was 62.5° against the mounted Cu(997) crystal, and the generated photoelectrons were captured at a normal angle (90°) by the 2-D detector of hemispherical electron analyzer (R4000 HiPP-3 APPES analyzer, Scienta Omicron). The clean Cu(997) surface was prepared by the same protocol with AP-STM and synchrotron-based AP-XPS measurements. The sample temperature was monitored using a directly connected K-type thermocouple attached to the edge of the polished crystal side. A home-built water trap was used for further purification of the high-purity $CO_2$ gas (99.999%) to prevent surface contamination by water impurities. All AP-XP spectra were collected after the AP-XPS system bakeout at 393 K for 120 hr. Details on the setup and its manipulation procedures were described in the previous work[66].

## Data availability

All data supporting the findings of this study are available within the article and/or the Supplementary Information. Source data are available with this manuscript. More detailed data are available from the corresponding author upon reasonable request. Source data are provided with this paper.

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

## Acknowledgements

We would like to acknowledge Dr. Miquel Salmeron (Lawrence Berkeley National Laboratory) for fruitful discussions. We also acknowledge the SOLEIL scientists for the provision of synchrotron radiation facilities at the beamline TEMPO. We thank the GIST-ESCA research group (Ms. Moonjung Jung, Mr. Dongwoo Kim, Mr. Minsik Seo, and Mr. Hyunsuk Shin) for their assistance with AP-XPS experiments. This work was supported by the National Research Foundation of Korea (NRF-2022R1A2C3004242, NRF-2022R1A2C2008448, NRF-2020K1A3A7A09080400, and NRF-2019K1A3A1A21030984) and the GIST Research Institute Grant funded by the Gwangju Institute of Science and Technology (GIST) 2023.

## Author contributions

J.K. and J.Y.P. conceived the project. J.K. performed AP-STM measurements. B.S.M. supervised synchrotron-based and lab-based AP-XPS measurements. J.K. and Y.Y. collected AP-XPS spectra. J.-J.G. and F.B. optimized optics parameters at the TEMPO beamline facility. T.W.G. prepared clean Cu single crystals for AP-STM and AP-XPS measurements. J.K., B.S.M., and J.Y.P. wrote the original manuscript. All authors discussed the experimental results for manuscript preparation.

## Competing interests

The authors declare no competing interests.
