## [Peer Review File · Nature Communications]

Revealing CO₂ dissociation pathways at vicinal copper (997) interfacesREVIEWER COMMENTS

Reviewer #1 (Remarks to the Author):

The STM part of this manuscript is of the highest quality. All the images shown in Figures S2-S7 and Figures 1-3 are remarkable. It is very difficult to resolve surfaces in the presence of gases like CO and CO₂ and the authors did a fantastic job both on the Cu(111) and Cu(997) surfaces. This collection of images is definitely one of the top I have seen in any HP-STM papers published so far. Only this is sufficient for the manuscript to be published at Nature Comm. I have two questions to the authors:

- 1- The authors present the effects of CO₂, CO, and O₂ on the atomic structure of Cu(997) separately but what is the conclusion? Is it CO or atomic O that causes the reconstructions in the steps during CO₂ dissociation, or is it somehow a combined effect of both (thus, neither test experiment results in what CO₂ by itself does)?
- 2- The XPS peak assignments are weird. CO at 531.2 eV is probably alright, but it is hard to imagine a CO peak below 530 eV. Moreover, there's no C 1s region shown. CO on Cu has one main and two satellite peaks in the C 1s region. The readers also have no information about how clean these samples are. Both of these are major issues that require the authors' attention.

Reviewer #2 (Remarks to the Author):

Kim et al. report on the activation of CO₂ over Cu(997) vicinal surfaces, using near-ambient pressure STM and XPS approaches. Having observed restructuring of the surface into nanoclusters that (at least partly) reverses on CO₂ removal, they then go on to consider how the surface changes during separate O₂ and CO exposures in order to understand the role that the different products of CO₂ dissociation play in catalyst restructuring.

Revealing the structural and chemical changes occurring at catalyst surfaces is extremely important in understanding the performance of catalysts during reactions and can inform the development of improved catalyst materials. In this work new understanding of how vicinal Cu surfaces reconstruct both irreversibly and reversibly during CO₂ exposure is introduced. I therefore found the study to be very interesting and an important contribution to the field, with the STM results particularly impressive. However, I believe the paper would benefit from some additional XPS/NEXAFS data and analysis to better support some of the claims that are mainly based on the real-space imaging with STM, but where chemical information is lacking. If the issues below can be addressed I believe the paper will be suitable for publication in Nature Communications:

- 1) XPS results are only presented for the CO exposure case, where based on the peak assignments made (see comment 2 about the need for further verification) their main role is to confirm CO absorption in different environments on the surface. XPS during the CO₂ exposure could be much more informative as this would indicate how much CO and O is adsorbed on the surface and thus the extent to which each is

responsible for the restructuring observed. This point is important to clarify, as CO and O have both been shown to lead to nanoclustering of the surface (here and in prior studies), and Ref 9 shows with XPS that it is the O which is responsible for clustering seen during CO₂ exposure. Whilst this may be different for Cu(997), it has not been clearly demonstrated which is responsible here.

2) XPS peak assignment is primarily based on a model assumed based on the STM results, but these assignments (particularly the 531.2 eV peak) are not well verified by any other means. This could be quite easily clarified by presenting similar data for the C1s region where corresponding features related to CO chemisorption would be expected to grow with exposure. Otherwise, the concern is that these might be some contaminant species such as hydroxide that might form due to displacement of water or other species from the chamber walls as the CO pressure is increased.

3) I also do not agree with the claim that “the recorded significant peak changes during AP-XPS measurements are far from the Cu oxidation features from water or oxygen contaminants.” Previous literature (e.g. refs 9, 16) indicates hydroxyl contaminants on Cu at 531.1-531.5 eV and adsorbed oxygen at 529.7-529.8 eV which could overlap closely with the 529.6 eV and 531.2 eV peaks reported here. Again this calls for measurement of the C1s region to clarify the origin of the peaks.

4) The manuscript states that “Further investigation in X-ray photoemission spectra for Cu 2p core-level state and Cu L_{2,3} absorption-edge indicates that the characterized metallic Cu surface (Cu⁰) does not transition to Cu₂O (Cu⁺) or CuO (Cu²⁺) with prolonged CO gas exposure (Supplementary Fig. 10).” In my view, Figure S10 shows a clear change between UHV and 0.2 mbar O₂ which is consistent with a transition toward Cu₂O. In fact the change is much greater than seen in the SI of ref 9 during CO₂ exposure at a similar pressure. Given the relatively large information depth of AEY NEXAFS (5-10 nm) this could correspond to quite significant Cu₂O formation at the very surface of the catalyst. This apparent oxidation of the Cu during CO exposure needs further consideration, particularly given the potential for oxidising contaminants as already highlighted in points 2 and 3. It would also be useful to compare to NEXAFS for the CO₂ exposed surface to determine the extent of any oxidation (see point 1).

Minor comments: The authors use both APPES and NAP-XPS interchangeably throughout the manuscript. I would suggest using a single-abbreviation throughout to avoid the impression that these are different techniques.

Some of the phrasing of the paper was not always clear, and may make the arguments presented harder to follow. I would recommend trying to ensure the text is as succinct and clear as possible. Some sentences where the meaning was not fully apparent include:

“However, the atomic O-covered Cu morphologies get involved in the irreversible reconstruction process. “

“We should address that CO-induced step-broken Cu clustering is not relevant to the gas contamination by nickel (Ni) carbonyl-containing compounds. Our AP-STM images highlight the reversible geometric

transition of the Cu(997) surface caused by the pressure gap effect, neither the step-oriented oxides by O220 nor contaminated Cu morphologies by Ni(CO)437 in the unpurified CO(g).”

“This attribute meets a surprising turn of events on the vicinal Cu surface at ambient pressure.”

The term “operando” is used at the start of the conclusion. This term is usually used to refer to measurement of a material system under realistic operating conditions e.g. those of a catalytic reaction. The room temperature conditions and ≤ 1 mbar pressures used herein are quite far from most realistic reactions, so I’m not convinced the term is appropriate here.

Reviewer #3 (Remarks to the Author):

In the manuscript entitled “Revealing CO₂ dissociation reaction at vicinal copper (997) interfaces” by Kim et al., the authors studied the structural morphology and chemical state of a vicinal Cu(997) surface under near-ambient CO₂ gas using AP-STM and AP-XPS. They found the formation of step-broken Cu nanoclusters on Cu(997) in 1 mbar CO₂, and the reversible behavior of Cu nanoclusters as a function of CO₂ gas pressure. Based on careful control experiments using CO(g) and O₂(g), the authors clarify the roles of dissociated CO and O species from CO₂ in the morphology change of the vicinal Cu surface.

The role of steps and kinks in the activation and reactions of CO₂ on Cu surfaces is of strong interest not only in academia but also to a wider audience in industry. The experiments were carried out very carefully (especially, checking the influence of Ni(CO)₄ clusters and residual H₂ gas), and the data presentation was of high quality. The reference to the previous literature is appropriate. However, some of data interpretation and conclusions require further clarifications with experimental evidence. Therefore, I would recommend its publication in Nature Communications after the following points are addressed by the authors.

1) The authors should show O 1s and C 1s AP-XPS spectra of Cu(997) in 1 mbar CO₂. In the present manuscript, the adsorption of CO₂ on Cu(997) was discussed and interpreted based solely on topographic images of AP-STM. Consequently, it is difficult to understand the chemical nature of the reconstructed vicinal Cu surface under near-ambient CO₂ gas. AP-XPS allows the quantification of surface chemical species. It is of particular interest to know the coverage of CO and O (and possibly CO₃) on Cu(997) in 1 mbar CO₂ and after evacuation. Note that the peak assignments of CO₂-related species were sometimes different among the previous publications. The authors are advised to check the experimental C/O atomic ratio from the C 1s/O 1s XPS peak intensities to confirm their peak assignments.

2) The novelty of the present work and the difference from the previous literatures should be explained more clearly. The combination of AP-XPS and AP-STM has been most successful in operando characterizations of single-crystal metal surfaces under ambient gas since the seminal work by F. Tao et

al. (Science 327, 850 (2010).). Since then, the formation of metal nanoclusters by adsorbates, step breaking, and its reversible behavior have been all reported in the literatures as cited by the authors in the manuscript. “What was the new physics found on the vicinal Cu(997) surface in the present study? Does the stepped surface just show higher reactivities compared to the flat surface?” These questions come partly from the lack of information about the chemical composition of the reconstructed Cu surface as explained in the comment #1. In this context, the authors should discuss the difference between the stepped Cu(997) surface and the flat Cu (100) and (111) surfaces (B. Eren et al., J. Am. Chem. Soc. 138, 8207 (2016).). On Cu (100) and (111) surfaces, CO desorbs from the surface and only O contributes to the surface restructuring. However, on Cu(997), both O and CO are involved in the morphology change.

3) The authors should present the corresponding C 1s AP-XPS spectra of Cu(997) in 0.2 mbar CO (Fig. 4) to justify the peak assignments in O 1s XPS spectra and evidence their claim of no CO dissociation. Three components of CO reflecting different adsorption sites should be observed in C 1s XPS spectra. The O 1s peak at 529.6 eV was assigned to CO at step edges, but it was close to the binding energies of atomic O on Cu surfaces: 529.7-529.8 eV on Cu(100) and Cu(111) (B. Eren et al., J. Am. Chem. Soc. 138, 8207 (2016).) and 529.5-529.9 eV on Cu(111) (K. Hayashida et al., ACS Omega 6, 26814 (2021).). The authors should mention that there is no contribution of atomic O in the 529.6 eV peak.

4) The authors are advised to define 1 ML on Cu(997) (e.g., 1 ML= XX atoms/cm²), and explain a coverage calibration procedure for CO (and CO₂) on Cu(997) at near-ambient conditions.

5) Please consider the following minor points:

5-1) (page 3) “under 0.2 and 0.8 mbar CO₂(g) conditions” should be “under 0.4 and 0.8 mbar CO₂(g) conditions”

5-2) (Fig. 1-4 captions) It is better to mention the temperature (300 K) of STM and XPS experiments explicitly.

5-3) (Fig. 2 caption) “Each white arrow” should be “Each green arrow”. In addition, please explain green dashed triangles in Fig. 2d.

Response to Reviewer #1:

General comment: *The STM part of this manuscript is of the highest quality. All the images shown in Figures S2-S7 and Figures 1-3 are remarkable. It is very difficult to resolve surfaces in the presence of gases like CO and CO₂ and the authors did a fantastic job both on the Cu(111) and Cu(997) surfaces. This collection of images is definitely one of the top I have seen in any HP-STM papers published so far. Only this is sufficient for the manuscript to be published at Nature Comm. I have two questions to the authors:*

Response: We appreciate the reviewer's positive evaluation and comments on our manuscript. We believe that our *in situ* observation images on the Cu(111) and Cu(997) surfaces could suggest a fundamental insight into unveiling the atomic-scale geometric structures at the gas-solid interface. Particularly, we would like to deliver a clue about the physicochemical property of vicinal Cu structures during the dissociation of CO₂ at ambient pressures. We hope our findings can adequately meet the readership and scientific interests in *Nature Communications*. We have prepared point-by-point responses for each question or suggestion by the reviewer. Please find the information below.

Reviewer comment #1-1: *1-The authors present the effects of CO₂, CO, and O₂ on the atomic structure of Cu(997) separately but what is the conclusion? Is it CO or atomic O that causes the reconstructions in the steps during CO₂ dissociation, or is it somehow a combined effect of both (thus, neither test experiment results in what CO₂ by itself does)?*

Response #1-1: We thank the reviewer for their constructive comments. Our scientific findings with the conclusion are addressed in the table of contents graphic. The catalytic dissociation of CO₂ produces atomic oxygen and molecular CO simultaneously, but the dissociated CO molecules have been considered negligible species, because the CO adsorption on the Cu surface is generally not preferred above room temperature. However, the CO adsorbate is able to reconstruct surface Cu atoms from the edge-sites of the metallic Cu surface, which would lead to wide surface restructuring even at a low CO coverage (~0.1 ML) at elevated CO pressures above 0.2 Torr CO(g) [B. Eren et al., *Science* **351**, 475-478 (2016), <https://doi.org/10.1126/science.aad8868>]. Furthermore, theoretical calculations suggested facile Cu atoms detachments from step-edges on the stepped Cu(100) surface [B. Eren et al., *Surf. Sci.* **651**, 210-214 (2016), <https://doi.org/10.1016/j.susc.2016.04.016>].

Based on previous reports, the geometric effect of the highly stepped Cu structure would affect energetics in overall catalytic reactions. The fundamental role of defect sites such as kinks and steps remains controversial, but their geometric structure alterations are very clear under oxidation conditions at the atomic level, as demonstrated by recently published real-time imaging results for Cu(100) and Cu(110) at the gas-solid interface [M. Li et al., *Nano Lett.* **22**, 1075-1082 (2022), <https://doi.org/10.1021/acs.nanolett.1c04124>].

In contrast, our fundamental knowledge has many gaps in molecular behaviors between CO adsorbates and step-edges on the Cu surface. In principle, the CO-induced surface restructuring of the stepped Cu surface differs from that of the stepped Pt surface, because the Cu catalyst is basically a soft material.

We could pick up useful insight into the randomly occurring surface restructuring at step-edges caused by repulsive CO-CO interactions on a Pt(557) substrate [F. Tao et al., *Science* **327**, 850-853 (2010), <https://doi.org/10.1126/science.1182122>; J. Kim et al., *JACS* **138**, 1110-1113; <https://doi.org/10.1021/jacs.5b10628>], but the physical property of the Cu surface is significantly different from that of the Pt surface: The π -electron back-donation between the CO adsorbate and metallic Cu [K. Hermann et al., *Phys. Rev. B* **35**, 9467, <https://doi.org/10.1103/PhysRevB.35.9467>] is much weaker than that of CO-Pt in theoretical models. Those atomic-scale interactions also affect the catalytic activity and selectivity of the CO₂ reduction reaction, and the catalytic reactions with atomic oxygen or CO adsorbate would proceed to different pathways depending on surface geometric and electronic structures.

Therefore, we need to investigate the atomic-scale interactions of atomic oxygen or CO adsorbates at the step-edge sites of the Cu surface. The vicinal Cu(997) surface is an appropriate model system to deal with this issue, in which its surface morphology structure offers very narrow saw-like geometries with a step-step distance of less than 2 nm. Our AP-STM observation images distinguish the trends of Cu atoms restructuring phenomena under O₂(g) or CO(g) conditions. In general, oxygen-induced surface reconstruction is irreversible. It is a well-established feature of Cu oxidation elsewhere, and the dissociated oxygen is highly adsorbed at the edge of steps until they are agglomerated with each other [T. J. Lawton et al., *J. Phys. Chem. C* **116**, 16054-16062 (2012), <https://doi.org/10.1021/jp303488t>].

In the present study, **Fig. 2a** also shows the reconstructed vicinal Cu surface that several local areas form oxygen-induced kinks and clusters at 300 K. Contrary to those images in **Figs. 2a-d**, the CO-induced surface restructuring on the Cu(997) surface could form relatively organized Cu clusters (average lateral size: ~2.1 nm), as shown in **Supplementary Fig. 6**; almost all the Cu clusters on each row of the stepped surface immediately disappeared after pumping down the reaction cell in UHV (**Figs. 2e and f**).

We find those unusual features with surface roughening and reversible Cu atoms restructuring simultaneously on the vicinal Cu(997) surface after introducing CO₂(g) at 300 K, as displayed in **Figs. 1c-f**. It means that the dissociated species from CO₂, both atomic oxygen and CO adsorbate, would interact with metallic Cu atoms at ambient pressures, which leads to the formation of more irregular kinks and Cu_x(CO)_y clusters along the direction of [1 -1 0] in the STM image.

We agree with the suggestion by reviewer that we should have addressed this argument more clearly in our manuscript, considering the broad readership in *Nature Communications*. In this revision process, we emphasized our findings of CO adsorbates- or atomic oxygen-induced surface restructuring phenomena and their correlations with CO₂ reduction reaction on the vicinal Cu(997) surface. We added more paragraphs to the Conclusion section, as follows:

1) On Page 18, Line 22:

The surface-molecule interaction of CO₂ at narrow-stepped Cu morphologies can effectively facilitate the dissociation of CO₂, producing atomic oxygen and CO adsorbates on the vicinal Cu surface at ambient pressure. The dissociated species can competitively occupy under-coordinated sites on the vicinal Cu(997) surface, and these molecular behaviors accompany the complicated surface restructuring phenomena with the formation of kinks and Cu nanoclusters around step-edge sites. In particular, CO adsorbates are able to propagate wide rearrangements of Cu atoms from the step-edge sites of the metallic Cu surface by repulsive CO–CO interactions at the gas-solid interface. Eventually, the clustering phenomenon happens widely across the Cu(997) surface, because the metallic Cu atoms could be lifted by CO adsorbates, but they are not fully ejected from the facet of vicinal Cu morphologies, due to the relatively weak binding strength between Cu and CO at 300 K. As a result, we find unusual reversible surface restructuring on the vicinal Cu surface as a function of adsorbed CO coverage. That is distinct from the oxygen-covered geometries, in that the oxidation process of the Cu surface is generally accomplished by irreversible surface reconstruction. Overall, the observed surface roughening trends caused by atomic oxygen and CO adsorbates may affect the result of successive catalytic reactions to produce critical intermediates, such as carbonate and formate, during the CO₂ reduction reaction. Indeed, those investigation results provide convincing evidence for the influence of surface morphology structures at the atomic level.

Reviewer comment #1-2: *2-The XPS peak assignments are weird. CO at 531.2 eV is probably alright, but it is hard to imagine a CO peak below 530 eV. Moreover, there's no C 1s region shown.*

CO on Cu has one main and two satellite peaks in the C 1s region. The readers also have no information about how clean these samples are. Both of these are major issues that require the authors' attention.

Response #1-2: We understand the reviewer's concern about our AP-XPS analysis results. To address these issues, we have significantly revised our technical interpretations of AP-XP spectra. We polished our arguments on CO-induced geometric alterations at the vicinal Cu interface in the aspect of the spectroscopic evidence. We added more AP-XPS data into the revised manuscript to specify the role of defect sites under gaseous CO conditions at 300 K. **Fig. 4** has been edited to clarify the observed physical meaning of the relationship between CO adsorbates and during the AP-XPS measurements.

First, we agree with the reviewer's comment on the CO-Cu species in C 1s core-level spectra. The major feature of CO adsorbates on the metallic surface could be generally found at 285–286 eV in the C 1s region; for instance, the adsorbed CO on the stepped Cu surfaces at certain CO coverages could be clearly characterized, as reported in previous literature for CO/Cu(211) [M. L. Ng et al., *Phys. Rev. Lett.* **114**, 246101 (2015), <https://doi.org/10.1103/PhysRevLett.114.246101>] and CO/Cu(997) [T. Koitaya et al., *J. Chem. Phys.* **144**, 054703 (2016), <https://doi.org/10.1063/1.4941060>] at low temperatures. The adsorbed CO molecules are usually observed with spectral complexes consisting of the main peak of CO-Cu and broad shape-up features at 286–293 eV in the C 1s core-level spectrum. We also detected the unique properties during AP-XPS measurements after introducing CO gas molecules into the analysis chamber. In our revised manuscript, we discussed the peak evolution trends of CO-Cu that appeared by careful analysis in C 1s spectra. Likewise, in our AP-STM observations, the observed properties of CO-Cu interactions at the vicinal Cu interface are specified in the revised manuscript compared with AP-XPS results of the CO/Cu(111) system.

Second, all AP-XPS experiments were conducted after multiple cycles of Ar⁺ ion-bombardment sputtering and annealing for cleaning the sample. We had a straightforward cleaning procedure in the same manner, as performed in our STM experiments. The original manuscript mentioned the detailed procedure in the Methods (*Sample preparation*) and Results (*Synchrotron-based AP-XPS analysis of CO-Cu species*) sections. In **Fig. 4a**, we already indicated that there is no detection of characteristic signals (bottom) from the as-cleaned Cu(997) surface in the O 1s core-level spectrum. In our revised manuscript, the cleanliness of our Cu crystal after cleaning cycles is confirmed again, as displayed in a C 1s core-level spectrum (**Supplementary Fig. 11**). The revised paragraphs and Supplementary Figures are as follows:

- 1) **Fig. 4 | Synchrotron-based AP-XPS measurements on the Cu(997) surface ($T = 300$ K).** (a) Collected O 1s core-level AP-XP spectra ($h\nu = 680$ eV) in UHV (bottom) and under CO gas conditions of 2×10^{-8} (middle) and 0.2 mbar (top). (b) Contour map analysis of time-lapse measurements for O 1s core-level spectra after filling the analysis chamber with CO gas ($P_{\text{CO}} = 0.2$ mbar). (c) Estimated relative CO(ads.) peak area ratio of $I_{\text{ads}}/I_{\text{gas}}$ plots from the time-lapse AP-XPS measurements.

- 2) **On Page 13, Line 20:**

The resolved peaks of 529.5 and 531.2 eV in the O 1s core-level spectrum are associated with the results of CO chemisorption on the vicinal Cu surface.

- 3) **On Page 14, Line 8:**

In contrast, we confirmed a clear rising trend of peak intensity at 531.2 eV and a peak shift of +0.1 eV from 529.5 eV during O 1s core-level measurements under the 0.2 mbar CO(g). The observed chemical species at 529.5 eV was not identified in O 1s core-level spectra of the Cu(111) surface in the same experimental environment (Supplementary Fig. 9). According to previous literature, the appeared peak or spectral shoulder at 529.5–529.6 eV would be assigned to surface oxygen by dissociation of CO or H₂O. It is reasonable that the introduced CO molecules may have interacted with a limited number of defect sites at the early stage of the overall chemical reaction over the vicinal Cu facets.

4) On Page 14, Line 17:

So, the CO chemisorption at metallic step-edge sites has relatively electron-rich states, depending on the CO coverage in XPS analysis.

5) On Page 15, Line 2:

The total amount of atomic oxygen species at 529.5–529.6 eV should be very small compared to the significant peak evolution at 531.2 eV, which is supported by the recorded survey spectrum indicating a substantial difference of peak intensity between Cu 2p and O 1s (Supplementary Fig. 10). The specified region of the peak evolution at 531.0–531.2 eV might also be attributed to the adsorbed water/hydroxyl species on the Cu surface, in accordance with previous reports. In general, molecular water does not prefer sticking onto the Cu terraces, which leads to thermodynamically metastable adsorption states on the Cu surface. The oxygen precovered Cu surface can provide much more favorable environments for molecular water adsorptions at a higher relative humidity (5%; T = 295 K), but our AP-XPS operating condition (P = 0.2 mbar CO; T = 300 K) is far from such a humid environment.

6) On Page 15, Line 12:

In fact, the characterized species at 531.2 eV could be originated from the roughened Cu surface by forming CO-Cu species; that physicochemical phenomenon was also clearly reported elsewhere by AP-XPS measurements under CO(g) conditions at room temperature. It was also possible to characterize a broadened spectral tail around 532.8 eV that originated from the CO chemisorption on kink and corner Cu atoms during the surface restructuring. Thus, the identified spectral features at 531.2 and 532.8 eV can be assigned to the chemisorbed CO in Fig. 4a, and we also find coexisting CO-Cu species at 285.6 and 287.7 eV in the C 1s core-level spectrum at that time (Supplementary Fig. 11). Based on interpretations of CO/Cu system in previous studies, it is rational that we assign the resolved peak at 531.2 eV as inner-sites of Cu step-edges on {997} facets, because the clustering phenomenon under ambient CO(g) environments could be related to the formation of CO-Cu species on the Cu surface.

Supplementary Fig. 11 | Collected C 1s core-level AP-XP spectra from the Cu(997) and Cu(111) surfaces ($h\nu = 400$ eV; T = 300 K). As-cleaned Cu(997) surface in UHV (bottom) shows no spectroscopic detection of chemical species. We find that multiple peaks appeared in the C 1s core-level spectra of Cu(997) and Cu(111) after introducing 0.2 mbar

CO(g) into the analysis chamber. The observed sp^2 (C=C) or sp^3 (C-C) carbon species at 284.2–284.3 eV would originate from the dissociation of CO or intrinsic carbon in bulk Cu lattice. The CO-induced morphology alterations on the vicinal Cu surface probably led to such upward segregation of buried carbon atoms at 300 K.

7) **On Page 15, Line 23:**

The geometric transition can affect the electronic structure and step-step interaction on the vicinal surface, which leads to the changes in energetics of CO chemisorption between flat and defective sites.

Response to Reviewer #2:

General comment: *Kim et al. report on the activation of CO₂ over Cu(997) vicinal surfaces, using near-ambient pressure STM and XPS approaches. Having observed restructuring of the surface into nanoclusters that (at least partly) reverses on CO₂ removal, they then go on to consider how the surface changes during separate O₂ and CO exposures in order to understand the role that the different products of CO₂ dissociation play in catalyst restructuring.*

*Revealing the structural and chemical changes occurring at catalyst surfaces is extremely important in understanding the performance of catalysts during reactions and can inform the development of improved catalyst materials. In this work new understanding of how vicinal Cu surfaces reconstruct both irreversibly and reversibly during CO₂ exposure is introduced. I therefore found the study to be very interesting and an important contribution to the field, with the STM results particularly impressive. However, I believe the paper would benefit from some additional XPS/NEXAFS data and analysis to better support some of the claims that are mainly based on the real-space imaging with STM, but where chemical information is lacking. If the issues below can be addressed I believe the paper will be suitable for publication in *Nature Communications*:*

Response: We appreciate the reviewer's statement that our manuscript could be suitable for publication in *Nature Communications* after addressing some issues. The development of rational catalysts for efficient CO₂ activation is currently a high priority in heterogeneous catalysis. CO₂ activation and its utilization to produce value-added chemical sources are directly correlated with the next-generation technology in sustainable clean energy conversion and climate change issues. Unfortunately, a lack of catalytic reaction pathways during the CO₂ activation progress at ambient pressures makes it difficult to improve the actual yield of the CO₂ reduction reaction. To tackle this challenging issue, we need to understand the complicated catalytic reaction steps of CO₂ activation in more detail at the atomic level. Moreover, those physicochemical behaviors must be characterized by nano-scale probing techniques under mimicked catalysis conditions. Within this line, the AP-STM is a suitable tool to observe atom-resolved surface morphologies as they are under reaction environments.

Our present study shows atomic-scale topographic images of critical evidence for reversible surface restructuring phenomena on the Cu(997) surface as a function of CO₂ pressures at ambient pressures. Unusual geometric features under 1 mbar CO₂(g), step-broken Cu clustering with reconstructed Cu step-edges, are consistently observed at the vicinal Cu interface, which

implies that the atomic-scale interactions of CO adsorbates and atomic oxygen may impact catalytic reactivity and selectivity in surface catalysis.

Synchrotron-based AP-XPS and AP-XAS are powerful spectroscopy techniques to identify chemical binding information during surface catalysis at ambient pressures. In particular, recently published literature revealing the pathways of CO₂ reduction reaction has contributed to establishing characterizations of intermediate species at the gas-solid interface [C. Heine et al., *JACS* **40**, 13246-13252 (2016), <https://doi.org/10.1021/jacs.6b06939>; R. M. Palomino et al., *J. Electron Spectrosc. Relat. Phenom.* **221**, 28-43 (2017), <https://doi.org/10.1016/j.elspec.2017.04.006>; L. Nguyen et al., *Chem. Rev.* **119**, 6822-6905 (2019), <https://doi.org/10.1021/acs.chemrev.8b00114>; X. Lian et al., *J. Phys. Chem. Lett.* **13**, 8264-8277 (2022), <https://doi.org/10.1021/acs.jpcllett.2c01191>]. That would extend to future studies on unraveling the CO₂ activation progress in electrocatalysis [Y. Yu et al., *PCCP* **16**, 11633-11639 (2014), <http://doi.org/10.1039/C4CP01054J>; A. Knop-Gericke et al., *J. Electron Spectrosc. Relat. Phenom.* **221**, 10-17 (2017), <https://doi.org/10.1016/j.elspec.2017.03.010>; J.-J. Velasco-Vélez et al., *J. Phys. D: Appl. Phys.* **54**, 124003 (2021), <https://doi.org/10.1088/1361-6463/abd2ed>].

We agree with the reviewer's constructive comments regarding our XPS/XAS data interpretations. Further spectroscopic analysis results and discussions will be helpful in understanding the disordered clustering phenomena observed on the Cu(997) surface. Basically, our manuscript focuses on the crucial role of the dissociated CO from CO₂ in the reversible nano-scale clustering behavior characterized by AP-STM. We believe that our real-time observation images of the step-broken clustering during the CO₂ dissociation would attract significant attention from a broad readership in *Nature Communications*. Because nobody had published such unexpectedly strong evidence for the reversible step-broken clustering on the Cu surface, our atom-resolved AP-STM images could be worthy of attention for further extensive discussions on the facile route of catalytic CO₂ dissociation at ambient pressures.

The oxygen-induced irreversible surface reconstruction of stepped Cu surfaces was already discussed elsewhere [A. Posada-Borbón et al., *Surf. Sci.* **675**, 64-69 (2018), <https://doi.org/10.1016/j.susc.2018.04.015>; B. Hagman et al., *Surf. Sci.* **715**, 121933 (2022), <https://doi.org/10.1016/j.susc.2021.121933>], and the characterization of dissociated atomic oxygen from CO₂ on model Cu surfaces has also been considerably repeated by employing the synchrotron-based AP-XPS technique at Advanced Light Source (United States) [B. Eren et al., *JACS* **138**, 8207-8211 (2016), <https://doi.org/10.1021/jacs.6b04039>; B. Hagman, *JACS* **140**, 12974-12979 (2018), <https://doi.org/10.1021/jacs.8b07906>] and Spring-8 (Japan) [T. Koitaya et al., *Top. Catal.* **59**, 526-531 (2016), <https://dx.doi.org/10.1007/s11244-015-0535-1>]. In the revised manuscript, we added more arguments and photoemission spectroscopy data to highlight the influence of CO

adsorbates at the gas-solid interface. Please find the detailed point-by-point responses to the reviewer's comments below.

Reviewer comment #2-1: *1) XPS results are only presented for the CO exposure case, where based on the peak assignments made (see comment 2 about the need for further verification) their main role is to confirm CO absorption in different environments on the surface. XPS during the CO₂ exposure could be much more informative as this would indicate how much CO and O is adsorbed on the surface and thus the extent to which each is responsible for the restructuring observed. This point is important to clarify, as CO and O have both been shown to lead to nanoclustering of the surface (here and in prior studies), and Ref 9 shows with XPS that it is the O which is responsible for clustering seen during CO₂ exposure. Whilst this may be different for Cu(997), it has not been clearly demonstrated which is responsible here.*

Response #2-1: The reviewer's suggestion that adding the spectroscopic analysis of the CO₂/Cu(997) is reasonable, because our manuscript covers the nano-scale clustering behaviors on the vicinal Cu surface under CO₂ dissociation conditions. As pointed out by the reviewer, the chemical binding data of atomic oxygen and CO would have been informative for our study, but they cannot provide evidence for the observed reversible step-broken clustering phenomenon directly, because the XPS technique has limited access to local morphology alterations within a few nanometer scales.

Typically, the irradiated X-ray beam spot has a size of about 50 μm × 50 μm–200 μm × 200 μm at the endstation of AP-XPS. In contrast, the observed morphology changes happen in the range of a few nanometers along the stepped Cu rows at the CO₂-Cu(997) interface. Another issue is that our experimental conditions of AP-STM (P_{CO₂} = 1 mbar; T = 300 K) still differ from the aggressive catalytic reaction conditions. It means the observed “slow” reaction kinetics is not enough to characterize such a dramatic change in micrometer-scale morphologies under the CO₂(g) condition.

In addition, it is possible to detect non-uniform structures using AP-XPS from the micrometer-scale cross-sectional area of the vicinal Cu surface. For example, even we can find a pronounced peak of the assigned atomic oxygen caused by the dissociation of CO₂ on the Cu(111) surface [B. Eren et al., *JACS* **138**, 8207-8211 (2016), <https://doi.org/10.1021/jacs.6b04039>]. The amount of dissociated adsorbates looks similar to that of the CO₂/Cu(997) surface [T. Koitaya et al., *Top. Catal.* **59**, 526-531 (2016), <https://dx.doi.org/10.1007/s11244-015-0535-1>]. We know it is hard to compare the published AP-XPS results directly to each other, because they were collected from different experimental conditions. Moreover, the characterized spot on the sample is too wide

to explain the physical meaning of atomic-scale restructuring of Cu atoms at ambient pressures. In other words, the averaged XPS signal collections from the surface are unsuitable for supporting the surface geometry changes at the atomic level.

So, we devised more control experiments which are focusing on the morphology alterations by the CO₂ dissociation. Our AP-STM images clearly demonstrate the different features of irreversible surface reconstruction by atomic oxygen at the step-edges and CO-induced reversible surface restructuring on the Cu(997) surface. Even though we find such complicated geometric alterations in AP-STM images of the CO₂/Cu(997) system, the observed reversible clustering phenomenon only happens under CO(g) conditions. The physical meaning of the reconstructed surface geometry at the step edge of Cu(997) could be explained by the influence of adsorbed atomic oxygen, which is consistent with previous literature and our control experiment results of the O₂/Cu(997) system.

Contrary to the measurements under CO₂(g) conditions, we can observe surface restructuring widely across the Cu(997) surface in CO(g) environments, due to the repulsive CO–CO interactions at the interface. We would like to suggest this point as the origin of the reversible surface clustering phenomenon. Unlike the CO₂/Cu(997) system, the CO-induced morphologic alteration is pretty straightforward, which would be enough to be probed with the AP-XPS technique.

Although the reviewer's comments make sense, we believe our logic also reasonably establishes the role of CO adsorbates on the Cu(997) surface. Indeed, the suggested reviewer's points should be considered when it comes to comparing the quantitative analysis in the collected core-level spectra. We have a future plan to characterize more details of the CO₂/Cu(997) system through multiple iterations of synchrotron-based experiments at different beam spot sizes, but we must discuss our strong evidence for the reversible surface restructuring phenomenon first in this study. We added more AP-XP spectra to support our arguments on the CO(g)/Cu(997) system to the revised manuscript below.

- 1) Supplementary Fig. 9** | The collected O 1s core-level spectra of Cu(111) surface under 0.2 mbar CO(g) conditions ($h\nu = 680$ eV; T = 300 K).

- 2) **Supplementary Fig. 10** | The collected survey spectrum on the Cu(997) surface under 0.2 mbar CO(g) ($h\nu = 1050 \text{ eV}$; $T = 300 \text{ K}$).

- 3) **Supplementary Fig. 11** | Collected C 1s core-level AP-XP spectra from the Cu(997) and Cu(111) surfaces ($h\nu = 400 \text{ eV}$; $T = 300 \text{ K}$). As-cleaned Cu(997) surface in UHV (bottom) shows no spectroscopic detection of chemical species. We find that multiple peaks appeared in the C 1s core-level spectra of Cu(997) and Cu(111) after introducing 0.2 mbar CO(g) into the analysis chamber. The observed sp^2 (C=C) or sp^3 (C-C) carbon species at

284.2–284.4 eV would originate from the dissociation of CO or intrinsic carbon in bulk Cu lattice. The CO-induced morphology alterations on the vicinal Cu surface probably led to this upward segregation of buried carbon atoms at 300 K.

4) On Page 14, Line 8:

In contrast, we confirmed a clear rising trend of peak intensity at 531.2 eV and a peak shift of +0.1 eV from 529.5 eV during O 1s core-level measurements under the 0.2 mbar CO(g). The observed chemical species at 529.5 eV was not identified in O 1s core-level spectra of the Cu(111) surface in the same experimental environment (Supplementary Fig. 9). According to previous literature, the appeared peak or spectral shoulder at 529.5–529.6 eV would be assigned to surface oxygen by dissociation of CO or H₂O. It is reasonable that the introduced CO molecules may have interacted with a limited number of defect sites at the early stage of the overall chemical reaction over the vicinal Cu facets.

5) On Page 15, Line 2:

The total amount of atomic oxygen species at 529.5–529.6 eV should be very small compared to the significant peak evolution at 531.2 eV, which is supported by the recorded

survey spectrum indicating a substantial difference of peak intensity between Cu 2p and O 1s (Supplementary Fig. 10).

Reviewer comment #2-2: *2)XPS peak assignment is primarily based on a model assumed based on the STM results, but these assignments (particularly the 531.2 eV peak) are not well verified by any other means. This could be quite easily clarified by presenting similar data for the C1s region where corresponding features related to CO chemisorption would be expected to grow with exposure. Otherwise, the concern is that these might be some contaminant species such as hydroxide that might form due to displacement of water or other species from the chamber walls as the CO pressure is increased.*

Response #2-2: As pointed out by the reviewer, we discussed the evolved peaks at 529.6, 531.2, and 532.8 eV in the O 1s core-level spectra with the observed phenomena by AP-STM measurements under CO(g) conditions. However, we also considered peak assignments from previous literature to analyze the detected features in our AP-XPS data. To our best knowledge, we found published papers from different research groups that claimed the peak at 531.0–531.2 eV in O 1s core-level spectra were related to the clustering feature of CO-Cu at ambient pressures [B. Eren et al., *Science* **351**, 475-478 (2016), <https://doi.org/10.1126/science.aad8868>; X. Zhang and S. Ptasinska, *ChemCatChem* **8**, 1632-1635 (2016), <https://doi.org/10.1002/cctc.201600046>]. They differ from CO adsorption on Cu surfaces with high coverage at low temperatures [M. L. Ng et al., *Phys. Rev. Lett.* **114**, 246101 (2015), <https://doi.org/10.1103/PhysRevLett.114.246101>; T. Koitaya et al., *J. Chem. Phys.* **144**, 054703 (2016), <https://doi.org/10.1063/1.4941060>].

We understand the concerns raised by the water/OH contamination issue during AP-XPS measurements. Recently published opinions also argue the controversial peak assignments caused by water contaminations on the metal or metal-oxide surfaces at ambient pressures [H. Idriss, *Surf. Sci.* **712**, 121894 (2021), <https://doi.org/10.1016/j.susc.2021.121894>; B. Eren et al., *Ambient Pressure Spectroscopy in Complex Chemical Environments Chapter 11*, 267-295 (2021), <https://doi.org/10.1021/bk-2021-1396.ch011>]. Despite the contamination issue, our discussions on the characterized feature at 531.2 eV could be reasonable, considering the previous literature and our supporting experimental results in the revised manuscript. It may be possible to see some overlapping features at around 531.0–531.4 eV, due to the unavoidable small water contamination at elevated pressures, but our AP-XPS results in C 1s core-level spectrum of Cu(997) (please see **Supplementary Fig. 11**) exhibit the adsorbed CO features at 285.6 and 287.7 eV (the main peak and its satellite). The spectral features were undiscerned on the Cu(111) surface under the same experimental conditions. Instead, we observed two peaks at 531.0 and 532.3 eV on the Cu(111)

surface at 0.2 mbar CO(g). Although the pronounced peak at 531.0 eV would also be related to the CO-Cu by surface restructuring, those observed features are unclear compared to that of the Cu(997). To address those scientific points, we have added discussions and revised several paragraphs, as follows:

1) **On Page 15, Line 5:**

The specified region of the peak evolution at 531.0–531.2 eV might also be attributed to the adsorbed water/hydroxyl species on the Cu surface, in accordance with previous reports. In general, molecular water does not prefer sticking onto the Cu terraces, which leads to thermodynamically metastable adsorption states on the Cu surface [B. J. Hinch and L. H. Dubois, *J. Chem. Phys.* **96**, 3262–3268 (1992), <https://doi.org/10.1063/1.461971>]. The oxygen precovered Cu surface can provide a much more favorable environment for molecular water adsorption at a higher relative humidity (5%; T = 295 K) [S. Yamamoto, *J. Phys. Condens. Matter* **20**, 184025 (2008), <https://doi.org/10.1088/0953-8984/20/18/184025>], but our AP-XPS operating condition (P = 0.2 mbar CO; T = 300 K) is far from such a humid environment.

2) **On Page 15, Line 12:**

In fact, the characterized species at 531.2 eV could be originated from the roughened Cu surface by forming CO-Cu species; that physicochemical phenomenon was also clearly reported elsewhere by AP-XPS measurements under CO(g) conditions at room temperature. It was also possible to characterize a broadened spectral tail around 532.8 eV that originated from the CO chemisorption on kink and corner Cu atoms during the surface restructuring. Thus, the identified spectral features at 531.2 and 532.8 eV can be assigned to the chemisorbed CO in **Fig. 4a**, and we also find coexisting CO-Cu species at 285.6 and 287.7 eV in the C 1s core-level spectrum at that time (Supplementary Fig. 11). Based on interpretations of CO/Cu system in previous reports, it is rational that we assign the resolved peak at 531.2 eV as inner-sites of Cu step-edges on {997} facets, because the clustering phenomenon under ambient CO(g) environments could be related to the formation of CO-Cu species on the Cu surface.

Reviewer comment #2-3: *3) I also do not agree with the claim that “the recorded significant peak changes during AP-XPS measurements are far from the Cu oxidation features from water or oxygen contaminants.” Previous literature (e.g. refs 9, 16) indicates hydroxyl contaminants on Cu at 531.1-531.5 eV and absorbed oxygen at 529.7-529.8 eV which could overlap closely with the 529.6*

eV and 531.2 eV peaks reported here. Again this calls for measurement of the C1s region to clarify the origin of the peaks.

Response #2-3: We understand the reviewer's concerns about our peak assignments and analysis in O 1s core-level spectra. Within the context, we revised several major points in our manuscript, including the issues of atomic O and hydroxyl group contaminations. Please see more details in **Response #2-2.**

Reviewer comment #2-4: *4)The manuscript states that “Further investigation in X-ray photoemission spectra for Cu 2p core-level state and Cu L_{2,3} absorption-edge indicates that the characterized metallic Cu surface (Cu⁰) does not transition to Cu₂O (Cu⁺) or CuO (Cu²⁺) with prolonged CO gas exposure (Supplementary Fig. 10).” In my view, Figure S10 shows a clear change between UHV and 0.2 mbar O₂ which is consistent with a transition toward Cu₂O. In fact the change is much greater than seen in the SI of ref 9 during CO₂ exposure at a similar pressure. Given the relatively large information depth of AEY NEXAFS (5-10 nm) this could correspond to quite significant Cu₂O formation at the very surface of the catalyst. This apparent oxidation of the Cu during CO exposure needs further consideration, particularly given the potentially for oxidising contaminants as already highlighted in points 2 and 3. It would also be useful to compare to NEXAFS for the CO₂ exposed surface to determine the extent of any oxidation (see point 1).*

Response #2-4: We can find previous literature dealing with Cu₂O (Cu⁺) or CuO (Cu²⁺) elsewhere, as indicated by the reviewer. We understand that the reviewer might have misunderstood the evolved minor feature at 946–947 eV in our Cu 2p core-level spectrum. As mentioned in the original manuscript, we could not detect any kinds of Cu oxidation features in further X-ray absorption spectroscopy measurements at ambient pressures (**Supplementary Fig. 13**). The spectral shape of characterized absorption edges was not much changed before and after introducing 0.2 mbar CO(g), and we could not see any other specific different features compared to the reference X-ray absorption data of the metallic and oxidized Cu surfaces [P. Jiang et al., *J. Chem. Phys.* **138**, 024704 (2013), <https://doi.org/10.1063/1.4773583>; M. C. Biesinger, *Surf. Interface Anal.* **49**, 1325–1334 (2017), <https://doi.org/10.1002/sia.6239>].

According to previous reports, the X-ray absorption measurements in Auger electron yield (AEY) mode support the very short escaping length of electrons, resulting in surface-sensitive characterizations with a probing depth of 1–2 nm [F. Lin et al., *Nat. Commun.* **5**, 3529 (2014), <https://doi.org/10.1038/ncomms4529>; F. Lin et al., *Chem. Rev.* **117**, 13123-13186 (2017), <https://doi.org/10.1021/acs.chemrev.7b00007>; J. W. Smith and R. J. Saykally, *Chem. Rev.* **117**,

13909-13934 (2017), <https://doi.org/10.1021/acs.chemrev.7b00213>; S.-M. Bak, *NPG Asia Mater.* **10**, 563-580 (2018), <https://doi.org/10.1038/s41427-018-0056-z>].

In our XAS measurements under 0.2 mbar CO(g) condition, only a limited amount of CO dissociation might be possible over the Cu(997) surface. But they are negligible, as shown in **Supplementary Fig. 10**; the relative peak intensity at 529.6 eV should be ultimately small compared to 531.2 eV. We cannot imagine any other oxygen source except for the negligible amount of atomic O from the CO dissociation. Also, the surface oxygen will not be incorporated into the Cu lattice structure. Our XAS data measured in the AEY mode is pretty surface-sensitive rather than that of characterizations in the total electron yield (TEY) mode (a probing depth: 5–10 nm). If the evolved minor species at 946–947 eV is an oxidized feature such as Cu₂O (Cu⁺) or CuO (Cu²⁺), it will also remain on the surface after gas evacuation. However, the detected minor feature disappeared immediately after CO gas evacuation from the analysis chamber (**Supplementary Fig. 14**).

The evolved small features in our Cu 2p core-level spectrum originated from the CO gas molecules at ambient pressures. It usually happens during the complicated processes of outgoing photoelectrons' energy losses [A. Jürgensen et al., *J. Electron Spectrosc. Relat. Phenom.* **232**, 111-120 (2019), <https://doi.org/10.1016/j.elspec.2018.11.004>] which are mainly involved in the metal-ligand charge transfers. The observed shake-up effect was also clearly characterized on the CO(g)/Cu(111) system elsewhere [X. Zhang and S. Ptasinska, *ChemCatChem* **8**, 1632-1635 (2016), <https://doi.org/10.1002/cctc.201600046>]. We added **Supplementary Fig. 14** to the revised manuscript to address this point as below.

- 1) **Supplementary Fig. 14** | Synchrotron-based X-ray photoelectron spectroscopy measurements for Cu 2p core-level states after gas evacuation ($h\nu = 1050$ eV; $T = 300$ K).

Reviewer comment #2-5: *Minor comments: The authors use both APPES and NAP-XPS interchangeably throughout the manuscript. I would suggest using a single-abbreviation throughout to avoid the impression that these are different techniques.*

Some of the phrasing of the paper was not always clear, and may make the arguments presented harder to follow. I would recommend trying to ensure the text is as succinct and clear as possible. Some sentences where the meaning was not fully apparent include:

“However, the atomic O-covered Cu morphologies get involved in the irreversible reconstruction process.”

“We should address that CO-induced step-broken Cu clustering is not relevant to the gas contamination by nickel (Ni) carbonyl-containing compounds. Our AP-STM images highlight the reversible geometric transition of the Cu(997) surface caused by the pressure gap effect, neither the step-oriented oxides by O₂²⁰ nor contaminated Cu morphologies by Ni(CO)₄³⁷ in the unpurified CO(g).”

“This attribute meets a surprising turn of events on the vicinal Cu surface at ambient pressure.”

Response #2-5: We appreciate the reviewer’s careful review of our manuscript. As suggested by the reviewer, we adopted a unified term of AP-XPS in the revised manuscript. We polished the indicated sentences, as requested by the reviewer. We hope to deliver the more clear messages to

the broad readership in *Nature Communications*. The revised sentences and paragraphs are as follows.

1) **On Page 21, Line 1:**

Synchrotron-based AP-XPS measurements. X-ray photoelectron spectroscopy experiments were carried out in the AP-XPS setup managed by the LCPMR group (Sorbonne Université), attached to the TEMPO beamline of the French national synchrotron facility (SOLEIL) in France.

2) **On Page 8, Line 10:**

The atomic O-covered Cu morphologies would be correlated with the irreversible reconstruction process at high atomic O coverages.

3) **On Page 12, Line 17:**

We note that the observed step-broken clustering is not involved in the gas contamination issue by nickel (Ni) carbonyl-containing compounds. Our AP-STM images demonstrate the reversible Cu(997) morphology alterations by repulsive CO–CO interactions. The observed clustering features under CO(g) conditions were not caused by well-known surface contaminations by step-oriented oxidation nor accumulated Ni(CO)₄ compounds.

4) **On Page 18, Line 11:**

Those unusual properties of CO-Cu interactions may cause the observed unexpected surface restructuring phenomena on the vicinal Cu surface as a function of CO pressures.

Reviewer comment #2-6: *The term “operando” is used at the start of the conclusion. This term is usually used to refer to measurement of a material system under realistic operating conditions e.g. those of a catalytic reaction. The room temperature conditions and ≤ 1 mbar pressures used herein are quite far from most realistic reactions, so I’m not convinced the term is appropriate here.*

Response #2-6: The reviewer’s comment is reasonable. Our experimental pressure and temperature conditions in AP-STM and AP-XPS measurements were almost fixed at 0.1–1 mbar and 300 K. Even though the pressure range is much higher ($> 10^{10}$ order of magnitude) than the conventional UHV experiments in surface science, our experimental environments still have a mechanistic gap ($> 10^3$ order of magnitude) compared to the industrial conditions. So, we revised the term of “operando” to “in situ” in our manuscript below.

1) On Page 1, Line 17:

Our *in situ* surface observations provide a more realistic insight into novel Cu nanocatalyst designs for efficient CO₂ conversion to renewable energy sources during C₁ chemical reactions.

2) On Page 18, Line 18:

Our atomic-scale *in situ* characterizations, combining AP-STM and synchrotron-based AP-XPS observations, provide convincing experimental information for the evolution of Cu nanoclusters on the Cu(997) surface by one-carbon molecules.

Response to the Reviewer #3:

General comment: *In the manuscript entitled “Revealing CO₂ dissociation reaction at vicinal copper (997) interfaces” by Kim et al., the authors studied the structural morphology and chemical state of a vicinal Cu(997) surface under near-ambient CO₂ gas using AP-STM and AP-XPS. They found the formation of step-broken Cu nanoclusters on Cu(997) in 1 mbar CO₂, and the reversible behavior of Cu nanoclusters as a function of CO₂ gas pressure. Based on careful control experiments using CO(g) and O₂(g), the authors clarify the roles of dissociated CO and O species from CO₂ in the morphology change of the vicinal Cu surface.*

The role of steps and kinks in the activation and reactions of CO₂ on Cu surfaces is of strong interest not only in academia but also to a wider audience in industry. The experiments were carried out very carefully (especially, checking the influence of Ni(CO)₄ clusters and residual H₂ gas), and the data presentation was of high quality. The reference to the previous literature is appropriate. However, some of data interpretation and conclusions require further clarifications with experimental evidence. Therefore, I would recommend its publication in Nature Communications after the following points are addressed by the authors.

Response: We appreciate the fair and very technical comments by the reviewer. As mentioned by the reviewer, unraveling the critical role of steps and kinds on Cu morphologies is very important to figure out surface catalysis and related intermediate formations during the CO₂ reduction reaction. So far, this issue has attracted much scientific interest from broad communities in academia and industry because enhancing the catalytic CO₂ activation process could be determined by geometric and electronic structures of Cu-based catalysts at ambient pressures.

Especially the steps and kinks on the Cu surface, those defect sites are deeply involved in unavoidable geometric alterations of Cu and Cu-based novel catalysts in redox conditions. For example, previous *in situ* transmission electron microscopy (TEM) studies indicate clearly that such a step edge-induced atomic-scale oxidation promotes the initial growth of oxide species by the modulation of gas-surface reaction kinetics on the Cu surface [G. Zhou et al., *Phys. Rev. Lett.* **109**, 235502 (2012), <https://doi.org/10.1103/PhysRevLett.109.235502>; L. Li et al., *Phys. Rev. Lett.* **113**, 136104 (2014), <https://doi.org/10.1103/PhysRevLett.113.136104>; M. Li et al., *Nat. Commun.* **12**, 2781 (2021), <https://doi.org/10.1038/s41467-021-23043-w>; M. Li et al., *Nano Lett.* **22**, 1075-1082 (2022), <https://doi.org/10.1021/acs.nanolett.1c04124>].

In the early step of catalytic CO₂ activation, the CO₂ dissociation [CO₂(g) → CO(ads.) + O(ads.)] over Cu catalysts could produce atomic oxygen on the Cu surface, which possibly interacts with step and kink sites spontaneously. Indeed, this simple catalytic reaction is correlated with

improvements in reactivity and selectivity for the CO₂ reduction reaction, in which even a limited number of sites are capable of catalytic propagation for CO₂ activation by tuning the activation energy barrier at the gas-solid interface. In contrast, the simultaneously produced CO(ads.) has been considered a negligible adsorbate, because the CO adsorption on the Cu surface is not preferred above room temperature. In addition, the removal of CO from Cu catalysts has been well-known as a way to prevent the CO-poisoning effect in real catalysis conditions [B. Eren et al., *JACS* **138**, 8207-8211 (2016), <https://doi.org/10.1021/jacs.6b04039>; B. Hagman, *JACS* **140**, 12974-12979 (2018), <https://doi.org/10.1021/jacs.8b07906>].

However, the interplay between CO adsorbates and the Cu surface would lead to an unexpected reconstruction of the metallic Cu surface by Cu atom detachments from step-edge sites at elevated CO pressure conditions [B. Eren et al., *Science* **351**, 475-478 (2016), <https://doi.org/10.1126/science.aad8868>; B. Eren et al., *Surf. Sci.* **651**, 210-214 (2016), <https://doi.org/10.1016/j.susc.2016.04.016>]. Even though the measured CO adsorbate coverage on the Cu(111) was low (~0.1 ML) in previous literature, the reported CO-induced surface reconstruction phenomena imply a low amount of CO could affect the alteration of surface geometry at ambient pressures. Also, DFT calculation results predicted that the CO-CO coupling is facile on Cu(100), Cu(111), and Cu(211). In particular, the Cu(100) has the lowest activation energy barrier (+0.45 eV), because the increased coverage of CO adsorbates and tensile strain on the Cu surface affects the reaction energetics at each facet, which would be involved in the activation condition of CO₂. [R. B. Sandberg et al., *Surf. Sci.* **654**, 56-62 (2016), <http://dx.doi.org/10.1016/j.susc.2016.08.006>]. Both experimental and theoretical results implicate that CO adsorbates would facilitate the rearrangements of Cu atoms even at low CO coverage.

Accordingly, the dissociated CO from CO₂ is no more than negligible adsorbate if the Cu surface consists of dense step-edge protrusions. We can see those specific morphology structures in size- and shape-controlled nanocatalysts, and active interactions between CO adsorbates and stepped Cu surface would lead to unexpected catalytic reaction pathways during the CO₂ reduction reaction. Yoshinobu and colleagues reported that the dissociation of CO₂ is facile on the vicinal Cu(997) surface, and their careful data analysis in AP-XP spectra supported evidence for the formation of carbonate species from the interaction of dissociated atomic oxygen with molecular CO₂ [T. Koitaya et al., *Top. Catal.* **59**, 526-531 (2016), <https://dx.doi.org/10.1007/s11244-015-0535-1>]. The reported CO₂/Cu(997) study emphasizes the important role of defect sites in CO₂ dissociation, albeit the analysis results assumed that the produced CO might be readily desorbed from the Cu surface at 340 K.

Our present study has focused on the CO-induced step-step interactions on the vicinal Cu surface at ambient pressures for the first time. Atom-resolved STM measurements clearly

demonstrate unusual reversible surface restructuring on the Cu(997) surface under 1 Torr CO₂(g) environment. In fact, we also find irreversible surface reconstruction in the measured same area due to the influence of dissociated atomic oxygen from CO₂. Control experiments clearly confirm those observed geometric alterations under CO or O₂ conditions, and we could exclude unwanted contamination effects caused by Ni(CO)₄ [L. Nguyen et al., *J. Phys. Chem. C* **117**, 971-977 (2013), <https://doi.org/10.1021/jp3086842>] or hydrogen [L. Österlund et al., *Phys. Rev. Lett.* **86**, 460 (2001), <https://doi.org/10.1103/PhysRevLett.86.460>] from our conclusion in AP-STM observations.

Nevertheless, identifying chemical states at the step-edge sites of the vicinal Cu surface is still challenging under ambient pressures. Contrary to real-space characterizations using AP-STM, we must consider various spectral features regarding chemical binding analysis in synchrotron-based photoelectron spectroscopy observations. Furthermore, the catalytic reaction system of CO₂/Cu(997) is not easy to understand using only the AP-XPS technique because of the gaseous CO₂ molecules' collision and scattering behaviors at the interface. They probably cause unpredictable physisorption or chemisorption under the X-ray radiation, resulting in obscure peak evolutions/overlapping in core-level XP spectra [M. Salmeron, *Top. Catal.* **61**, 2044-2051 (2018), <https://doi.org/10.1007/s11244-018-1069-0>].

Thus, we understand the reviewer's concern about the clarifications with our experimental analysis. To clarify our conclusion, we have supported more spectroscopic analysis results in the revised manuscript, as suggested by the reviewer. Please see below our point-by-point responses to the reviewer's comments.

Reviewer comment #3-1: *1) The authors should show O 1s and C 1s AP-XPS spectra of Cu(997) in 1 mbar CO₂. In the present manuscript, the adsorption of CO₂ on Cu(997) was discussed and interpreted based solely on topographic images of AP-STM. Consequently, it is difficult to understand the chemical nature of the reconstructed vicinal Cu surface under near-ambient CO₂ gas. AP-XPS allows the quantification of surface chemical species. It is of particular interest to know the coverage of CO and O (and possibly CO₃) on Cu(997) in 1 mbar CO₂ and after evacuation. Note that the peak assignments of CO₂-related species were sometimes different among the previous publications. The authors are advised to check the experimental C/O atomic ratio from the C 1s/O 1s XPS peak intensities to confirm their peak assignments.*

Response #3-1: We thank the detailed technical comments by the reviewer. As pointed out by the reviewer, we considered peak assignments for the detected species in our AP-XP spectra based on our AP-STM observation results. Even though the topographic images cannot clarify the exact

chemical binding information, the identified CO chemisorption features on the Cu(997) surface support critical evidence for the CO-induced surface restructuring under CO(g) conditions. Compared with imaging techniques, the synchrotron-based AP-XPS technique is able to provide physisorption and chemisorption species on the surface at ambient pressures. The obtained qualitative/quantitative chemical information is very useful in characterizing catalytic behaviors under the reaction condition, which applies equally to our present study.

As suggested by the reviewer, we added the experimental assignment of the C/O atomic ratio to the revised manuscript. To address this point, we followed an analysis procedure from the literature [T. Koitaya et al., *Top. Catal.* **59**, 526-531 (2016), <https://doi.org/10.1007/s11244-015-0535-1>]. We normalized peak intensities of both O 1s and C 1s core-level spectra against the commonly observed gaseous CO peak to minimize the photoelectron diffraction effect and the influence of the analyzer transmission function at different photon kinetic energies. This procedure is pretty useful to interpret chemical species empirically, albeit we cannot thoroughly exclude the various effects caused by the different core-electron ionized cross-sections between O 1s and C 1s core-level spectra. Even so, the interpretation by following this procedure could provide a relative atomic ratio of O/C. The revised paragraphs in our revised manuscript as below.

1) **On Page 16, Line 20:**

We estimated the composition ratio of O/C from the integrated peak areas for assigned CO species in O 1s and C 1s core-level spectra at 0.2 mbar CO(g). After normalizations of the AP-XP spectra by peak intensities of CO(g), we compared the integrated peak areas of assigned CO-Cu species in O 1s (531.2 and 532.8 eV) and C 1s (285.6 and 287.7 eV) spectra. In this quantitative analysis, we obtained a reliable value of 1.1 ± 0.1 for the atomic ratio of O/C, which confirms the peak evolutions by molecular interactions of CO adsorbates on the Cu(997) surface. Technically, we cannot thoroughly exclude the influence of the photoelectron diffraction effect and differences in core-level ionization cross-section between O 1s and C 1s spectra at different photon kinetic energies. Despite those experimental uncertainties, the empirical analysis method is useful for estimating the relative compositional ratio of O/C from the collected core-level spectra.

Reviewer comment #3-2: *2) The novelty of the present work and the difference from the previous literatures should be explained more clearly. The combination of AP-XPS and AP-STM has been most successful in operando characterizations of single-crystal metal surfaces under ambient gas since the seminal work by F. Tao et al. (Science 327, 850 (2010)). Since then, the formation of metal nanoclusters by adsorbates, step breaking, and its reversible behavior have been all reported*

*in the literatures as cited by the authors in the manuscript. “What was the new physics found on the vicinal Cu(997) surface in the present study? Does the stepped surface just show higher reactivities compared to the flat surface?” These questions come partly from the lack of information about the chemical composition of the reconstructed Cu surface as explained in the comment #1. In this context, the authors should discuss the difference between the stepped Cu(997) surface and the flat Cu (100) and (111) surfaces (B. Eren et al., *J. Am. Chem. Soc.* **138**, 8207 (2016).). On Cu (100) and (111) surfaces, CO desorbs from the surface and only O contributes to the surface restructuring. However, on Cu(997), both O and CO are involved in the morphology change.*

Response #3-2: We thank the reviewer’s constructive comments on our present study. As mentioned by the reviewer, the representative study of step-broken Pt clustering on the Pt(557) by Somorjai and colleagues showed that CO adsorbates could induce unexpected step-step interactions by high CO coverages at ambient pressures. We also revealed the geometric alterations of CO-induced Pt nanoclusters on the Pt(557) at elevated temperature conditions [J. Kim et al., *JACS* **138**, 1110-1113 (2016), <https://doi.org/10.1021/jacs.5b10628>]. To emphasize the novelty of Cu clustering in our present study, we highlighted different physical properties of Cu compared to Pt in our manuscript. The commonly observed surface restructuring phenomena on Pt(557) and Cu(997) look similar to each other, but their mechanistic behaviors must be different, as mentioned in our manuscript “*Of course, the chemical binding strength of CO-Cu is much weaker than that of CO-Pt, on account of insufficient π -electron back-donation from the metallic Cu to the CO adsorbate*”.

In fact, as shown in **Fig. 1**, we observed reversible and irreversible surface restructuring features on the Cu(997) surface after introducing CO₂(g) at ambient pressures. To our best knowledge, the characterized unusual phenomena were not published elsewhere before; our atom-resolved AP-STM images provide fruitful information on the CO₂ activation progress, which was also compared with O- or CO-induced surface morphologies on the Cu(997) surface. In principle, the dissociation of CO₂ generates atomic O and CO adsorbate simultaneously at the vicinal Cu interface. The observed CO₂ dissociation on the Cu(997) is much more active than the Cu(111) results, as indicated by the reviewer. However, our manuscript focuses on revealing the atomic-scale morphologic alterations during the CO₂ dissociation at the vicinal Cu interface, not reaction yields.

Our AP-STM images suggest strong evidence for the restructuring phenomena of Cu atoms caused by the interplay between atomic O and CO adsorbate. The control experiments have confirmed the irreversibly or reversibly occurring surface restructuring behaviors under the O₂(g)

or CO(g) environments, and we specified their critical roles in catalytic CO₂ activation at ambient pressures. To address our arguments on the observed unusual phenomena, we polished our conclusion section in the revised manuscript below.

1) On Page 18, Line 22:

The surface-molecule interaction of CO₂ at narrow-stepped Cu morphologies can effectively facilitate the dissociation of CO₂, producing atomic oxygen and CO adsorbates on the vicinal Cu surface at ambient pressure. The dissociated species can competitively occupy under-coordinated sites on the vicinal Cu(997) surface, and these molecular behaviors accompany the complicated surface restructuring phenomena with the formation of kinks and Cu nanoclusters around step-edge sites. In particular, CO adsorbates are able to propagate wide rearrangements of Cu atoms from the step-edge sites of the metallic Cu surface by repulsive CO–CO interactions at the gas-solid interface. Eventually, the clustering phenomenon happens widely across the Cu(997) surface, because the metallic Cu atoms could be lifted by CO adsorbates, but they are not fully ejected from the facet of vicinal Cu morphologies, due to the relatively weak binding strength between Cu and CO at 300 K. As a result, we find unusual reversible surface restructuring on the vicinal Cu surface as a function of adsorbed CO coverage. That is distinct from the oxygen-covered geometries, in that the oxidation process of the Cu surface is generally accomplished by irreversible surface reconstruction. Overall, the observed surface roughening trends caused by atomic oxygen and CO adsorbates may affect the result of successive catalytic reactions to produce critical intermediates, such as carbonate and formate, during the CO₂ reduction reaction. Indeed, those investigation results provide convincing evidence for the influence of surface morphology structures at the atomic level.

Reviewer comment #3-3: *3) The authors should present the corresponding C 1s AP-XPS spectra of Cu(997) in 0.2 mbar CO (Fig. 4) to justify the peak assignments in O 1s XPS spectra and evidence their claim of no CO dissociation. Three components of CO reflecting different adsorption sites should be observed in C 1s XPS spectra. The O 1s peak at 529.6 eV was assigned to CO at step edges, but it was close to the binding energies of atomic O on Cu surfaces: 529.7-529.8 eV on Cu(100) and Cu(111) (B. Eren et al., J. Am. Chem. Soc. 138, 8207 (2016).) and 529.5-529.9 eV on Cu(111) (K. Hayashida et al., ACS Omega 6, 26814 (2021).). The authors should mention that there is no contribution of atomic O in the 529.6 eV peak.*

Response #3-3: We appreciate the reviewer's comments regarding the peak assignments in O 1s core-level spectra under CO(g) conditions. As indicated by the reviewer, the detected feature at 529.5–530.1 eV could be assigned as atomic oxygen. Technically, the peak deconvolution analysis to determine the amount of adsorbed CO and dissociated atomic oxygen has many uncertainties, because we do not have more detailed chemical binding information on at least three different photon kinetic energies for O 1s and C 1s core-level spectra. We admit that the possibility of CO dissociation could not be entirely excluded from our data. This point is also important to interpret the physical meaning of appeared peaks at 531.0–532.8 eV in O 1s core-level spectra. In our revised manuscript, we added the corresponding C 1s core-level spectra of Cu(997) under 0.2 mbar CO(g) and more discussions to assign the identified chemisorption species as follows.

1) **On Page 14, Line 2:**

Fig. 4 | Synchrotron-based AP-XPS measurements on the Cu(997) surface (T = 300 K). (a) Collected O 1s core-level AP-XP spectra ($h\nu = 680$ eV) in UHV (bottom) and under CO gas conditions of 2×10^{-8} (middle) and 0.2 mbar (top). (b) Contour map analysis of time-lapse measurements for O 1s core-level spectra after filling the analysis chamber with CO gas ($P_{\text{CO}} = 0.2$ mbar). (c) Estimated relative peak area ratio of $I_{\text{ads}}/I_{\text{gas}}$ from the time-lapse AP-XPS measurements.

2) **Supplementary Fig. 11 |** Collected C 1s core-level AP-XP spectra from the Cu(997) and Cu(111) surfaces ($h\nu = 400$ eV; T = 300 K). As-cleaned Cu(997) surface in UHV (bottom) shows no spectroscopic detection of chemical species. We find that multiple peaks appeared in the C 1s core-level spectra of Cu(997) and Cu(111) after introducing 0.2 mbar CO(g) into the analysis chamber. The observed sp^2 (C=C) or sp^3 (C-C) carbon species at

284.2–284.3 eV would originate from the dissociation of CO or intrinsic carbon in bulk Cu lattice. The CO-induced morphology alterations on the vicinal Cu surface probably led to such upward segregation of buried carbon atoms at 300 K.

3) On Page 14, Line 11:

According to previous literature, the appeared peak or spectral shoulder at 529.5–529.6 eV would be assigned to surface oxygen by dissociation of CO or H₂O. It is reasonable that the introduced CO molecules may have interacted with a limited number of defect sites at the early stage of the overall chemical reaction over the vicinal Cu facets.

4) On Page 15, Line 5:

The specified region of the peak evolution at 531.0–531.2 eV might also be attributed to the adsorbed water/hydroxyl species on the Cu surface, in accordance with previous reports. In general, molecular water does not prefer sticking onto the Cu terraces, which leads to thermodynamically metastable adsorption states on the Cu surface. The oxygen precovered Cu surface can provide much more favorable environment for molecular water adsorption at a higher relative humidity (5%; $T = 295$ K), but our AP-XPS operating condition ($P =$

0.2 mbar CO; T = 300 K) is far from such a humid environment.

5) On Page 15, Line 12:

In fact, the characterized species at 531.2 eV could be originated from the roughened Cu surface by forming CO-Cu species; that physicochemical phenomenon was also clearly reported elsewhere by AP-XPS measurements under CO(g) conditions at room temperature. It was also possible to characterize a broadened spectral tail around 532.8 eV that originated from the CO chemisorption on kink and corner Cu atoms during the surface restructuring. Thus, the identified spectral features at 531.2 and 532.8 eV can be assigned to the chemisorbed CO in **Fig. 4a**, and we also find coexisting CO-Cu species at 285.6 and 287.7 eV in the C 1s core-level spectrum at that time (**Supplementary Fig. 11**). Based on interpretations of CO/Cu system in previous reports, it is rational that we assign the resolved peak at 531.2 eV as inner-sites of Cu step-edges on {997} facets, because the clustering phenomenon under ambient CO(g) environments could be related to the formation of CO-Cu species on the Cu surface.

Reviewer comment #3-4: *4) The authors are advised to define 1 ML on Cu(997) (e.g., 1 ML= XX atoms/cm²), and explain a coverage calibration procedure for CO (and CO₂) on Cu(997) at near-ambient conditions.*

Response #3-4: We thank the reviewer for their technical advice in defining the coverage of CO on the Cu(997) surface. In principle, the adsorbate coverage can be determined by using an atomic-scale model structure with the assumption of steady-state equilibrium conditions [B. Eren et al., *J. Phys. Chem. C* **119**, 14669-14674 (2015), <https://doi.org/10.1021/jp512831f>]. Even though it is hard to define a certain model structure in our present study, because CO adsorbates induce surface restructuring at ambient pressures, we added a defined coverage from the model structure [T. Koitaya et al., *J. Chem. Phys.* **144**, 054703 (2016), <https://doi.org/10.1063/1.4941060>] to compare our estimated CO coverages from AP-STM images.

1) On Page 21, Line 22:

The saturation coverage of CO(ads.) was estimated from the (1.4 × 1.4)-CO superstructure on the Cu(997) surface ($\theta_{\text{CO}} = 0.52$ molecules/one surface Cu atom).

Reviewer comment #3-5: *5) Please consider the following minor points:*

5-1) (page 3) “under 0.2 and 0.8 mbar CO₂(g) conditions” should be “under 0.4 and 0.8 mbar CO₂(g) conditions”

5-2) (Fig. 1-4 captions) It is better to mention the temperature (300 K) of STM and XPS experiments explicitly.

5-3) (Fig. 2 caption) “Each white arrow” should be “Each green arrow”. In addition, please explain green dashed triangles in Fig. 2d.

Response #3-5: We thank the reviewer for their suggestions and have revised our manuscript, as shown below:

1) On Page 2, Line 19:

Meanwhile, synchrotron-based ambient pressure X-ray photoelectron spectroscopy (AP-XPS) indicated that the stepped Cu surface of Cu(100) and Cu(997) could allow the dissociation of CO₂ at step sites under 0.4 and 0.8 mbar CO₂(g) conditions, despite the low sticking probability of CO₂.

2) On Page 4, Line 11:

Fig. 1 | The ideal Cu(997) surface structure and observed morphology alterations under 1 mbar CO₂(g) (T = 300 K).

3) On Page 6, Line 8:

Each green arrow on the STM image represents a row of dissociatively adsorbed oxygen at the step-edge site.

4) On Page 6, Line 9:

The representative dotted green triangles on the STM image indicate the evolved CuO_x clusters after O₂ gas exposures at 500 K. Both topographic images were recorded at 300 K.

5) On Page 11, Line 2:

Fig. 3 | Time-lapse morphology observations on the Cu(997) surface under 1 mbar CO gas conditions (T = 300 K).

6) On Page 14, Line 2:

Fig. 4 | Synchrotron-based AP-XPS measurements on the Cu(997) surface (T = 300 K).

REVIEWER COMMENTS

Reviewer #1 (Remarks to the Author):

Both of the points I raised in the previous round of review are nicely answered. The changes done by the authors are also adequate. I think the interpretation of the APXPS data in this version is better and more correct than the previous version. The authors also clearly point out why they think Cu is reconstructing. As I mentioned in the previous round of review, the APSTM part of this manuscript is also top notch. The work is original in its own right: So far our knowledge in restructuring of Cu surfaces in the presence of CO₂ or CO at ambient pressures is limited to low-index surfaces. This study extends this to the vicinal surfaces. It turns out that there are similarities between the behavior of low-index and vicinal surfaces. Chiefly among the latter is the clusters formed by adsorption of CO or CO₂ gases are confined to the narrow terraces for the vicinal surfaces.

I should mention that obtaining atomically resolved images with APSTM is a daunting task. The authors did a fantastic job in this article. I recommend the publication of the article in its current form.

Reviewer #2 (Remarks to the Author):

A version of these comments with figures is attached as a PDF.

The authors have added some extra arguments and data in response to my previous questions. On point #2-1, #2-5, and #2-6 they have largely addressed my concerns, however the additional data added and their replies to points #2-2, #2-3, #2-4, have deepened my concerns about significant contamination in the APXPS measurements following CO dosing. In my view the large contributions from adventitious carbon (and likely H₂O) contamination make firm assignments of the CO species unreliable, and there is strong evidence of oxidation of the Cu surface during CO exposure further supporting a contamination issue. The STM study presented by the authors is high quality, however at present I don't think the conclusions made concerning the APXPS/NEXAFS measurements are adequately supported by the data presented. I would therefore suggest the authors either repeat the APXPS/NEXAFS measurements taking care to minimize these contamination sources, or consider resubmitting without the APXPS part perhaps with further chemical evidence from another technique.

I provide my detailed responses to each point below.

Reviewer comment #2-1:

Whilst I think APXPS studies with CO₂ exposure would strengthen the paper (as does reviewer 3), I accept the authors' arguments about the reversibility of the clustering induced by CO being observed by STM.

Reviewer comment #2-2:

The inclusion of the C1s data now highlights a significant issue with contamination of the surface with adventitious carbon (labelled C*) when CO is dosed. This is seen to dominate on both Cu(111) and Cu(997) now shown in Figure S11. This could include oxygenated hydrocarbons, confusing the interpretation of the O1s, and also the strong overlap of adventitious carbon likely influences the attempts at corroborating CO adsorption by comparing the C1s and O1s intensities. In fact, I was quite surprised by the authors response to reviewer point 3-1: "In this quantitative analysis, we obtained a reliable value of 1.1 ± 0.1 for the atomic ratio of O/C, which confirms the peak evolutions by molecular interactions of CO adsorbates on the Cu(997) surface." I can't see that this can be reliable with the extent of overlap with the adventitious carbon, and this issue didn't seem to be mentioned.

Reviewer comment #2-3:

The authors have made some alterations to the manuscript to acknowledge atomic O and hydroxyl group contamination, but they still describe these as small in their response to comment #2-2 and the changes made to the manuscript. However the spectrum the present in Figure 4a of the manuscript has very similar features and intensity ratios to experiments with water adsorption shown in ref 28 at 0.05 torr (except for the H₂O gas phase species isn't present) and included here as a figure for convenience. Given the extent of contamination noted in #2-2 above, and the surface oxidation of Cu noted in #2-4 below, this all points to significant surface contamination when the CO is introduced.

Reviewer comment #2-4:

The authors have not fully addressed the issue I was trying to refer to, they discuss the Cu2p spectra, but it is well established that Cu⁰ and Cu⁺ are not readily distinguished by measurements of the Cu2p regions, so comparisons of this are not really relevant. I was instead referring to the NEXAFS, and therefore realize my comment may have been ambiguous (including some small typographical errors). I therefore restate my point here more clearly:

The Cu L-edge NEXAFS in Figure S13 (right hand panel) in the revised manuscript shows a clear change between UHV and 0.2 mbar CO which is consistent with a transition toward Cu₂O. In fact the change is much greater than seen in the Supplementary Information of ref 16 during CO₂ exposure at a similar pressure, where O is expected to be supplied by CO₂ dissociation.

Below is a comparison of this, where I have shifted one of the authors spectra so they are overlaid for more clear comparison (left hand panel), and included the data from ref 16 (right hand panel).

Based on a rough measurement of the spectra in the current work, the intensity difference between the first peak and trough is ~ 1.4 x larger when the CO is added compared to under UHV. For reference 16, the increase is only 1.05x following CO₂ exposure. Both measurements are AEY-NEXAFS so arguments about different depth sensitivities are not relevant. It should also be noted that the L₃ edge position is also shifted to slightly higher energy after the CO exposure in the current work. This indicates oxidation of the Cu surface to form Cu₂O, however the authors do not account for this and suggest that they "cannot imagine any other oxygen source except for the negligible amount of atomic O from the CO

dissociation.” I agree that for a clean chamber and CO gas line, Cu oxidation is not expected given CO is highly reducing and should in fact remove oxygen from the Cu surface. Instead, the fact they do see oxidation suggests a significant source of contamination, likely water or oxygenated hydrocarbons being displaced from the walls of the chamber/gas line. This is further supported by the significant adventitious carbon they see appear in the C1s region. This calls into question much of the APXPS results within this manuscript, as H₂O, CH_x, and CH_xO_y are likely to overlap with many of the features being attributed.

Reviewer comment #2-5, 2-6: These points have been adequately addressed.

Reviewer #3 (Remarks to the Author):

I appreciate the authors' effort to reply to my comments and revise the manuscript. I am satisfied with the responses #3-3, 3-4, 3-5, but not with the responses #3-1, 3-2. Especially, in the response #3-1, the authors chose not to show O 1s and C 1s AP-XPS spectra of Cu(997) in 1 mbar CO₂ in the revised manuscript, which were important to support their conclusion and prove the novelty of the work as discussed below. Therefore, I could not recommend its publication in Nature Communications in the current form.

The authors concluded that the surface morphology change of Cu(997) in near-ambient CO₂ gas originates mainly from CO adsorbates, forming Cu_x(CO)_y clusters; atomic O atoms also have small contributions to it. This conclusion is based solely on AP-STM topographic images that show the reversible and irreversible behaviors of the Cu(997) surface in CO and O₂, respectively. I think that this is not strong enough to support their conclusion and additional spectroscopic evidence by O 1s and C 1s AP-XPS spectra of Cu(997) in 1 mbar CO₂ is necessary. In addition, the chemical nature of the reconstructed Cu(997) surface is very important to prove the novelty of the present study. In my opinion, the formation of metal nanoclusters, step breaking, and its reversible behavior by one type of adsorbates have been all reported in the literature (e.g., CO/Pt(557), Science 327, 850 (2010).) and lack in novelty. The novelty of this work could be that the surface morphology change is driven by two or more types of adsorbates. Therefore, it is particularly important to know how different surface adsorbates of CO and O (and possibly CO₃) compete or cooperate to reconstruct the surface using AP-XPS.

Response to Reviewer #1:

General comment: *Both of the points I raised in the previous round of review are nicely answered. The changes done by the authors are also adequate. I think the interpretation of the APXPS data in this version is better and more correct than the previous version. The authors also clearly point out why they think Cu is reconstructing. As I mentioned in the previous round of review, the APSTM part of this manuscript is also top notch. The work is original in its own right: So far our knowledge in restructuring of Cu surfaces in the presence of CO₂ or CO at ambient pressures is limited to low-index surfaces. This study extends this to the vicinal surfaces. It turns out that there are similarities between the behavior of low-index and vicinal surfaces. Chiefly among the latter is the clusters formed by adsorption of CO or CO₂ gases are confined to the narrow terraces for the vicinal surfaces.*

I should mention that obtaining atomically resolved images with APSTM is a daunting task. The authors did a fantastic job in this article. I recommend the publication of the article in its current form.

Response: We are extremely grateful for the reviewer's positive feedback and recommendation to accept the paper in its current form. Our present study provides experimental evidence for vicinal Cu surface restructuring under CO₂(g) or CO(g) environments at the atomic level. As mentioned by the reviewer, the novelty of this work is in extending our insight from the well-established low-index surfaces into the complicated vicinal Cu surfaces at ambient pressures. We also believe that our *in situ* observation results contribute to understanding the relationship between catalytic CO₂ dissociation pathways and narrowed surface geometries within a 1–2 nm scale at the gas-solid interface.

Response to Reviewer #2:

General comment: *A version of these comments with figures is attached as a PDF. The authors have added some extra arguments and data in response to my previous questions. On point #2-1, #2-5, and #2-6 they have largely addressed my concerns, however the additional data added and their replies to points #2-2, #2-3, #2-4, have deepened my concerns about significant contamination in the APXPS measurements following CO dosing. In my view the large contributions from adventitious carbon (and likely H₂O) contamination make firm assignments of the CO species unreliable, and there is strong evidence of oxidation of the Cu surface during CO exposure further supporting a contamination issue. The STM study presented by the authors is high quality, however at present I don't think the conclusions made concerning the APXPS/NEXAFS measurements are adequately supported by the data presented. I would therefore suggest the authors either repeat the APXPS/NEXAFS measurements taking care to minimize these contamination sources, or consider resubmitting without the APXPS part perhaps with further chemical evidence from another technique. I provide my detailed responses to each point below.*

The inclusion of the C1s data now highlights a significant issue with contamination of the surface with adventitious carbon (labelled C) when CO is dosed. This is seen to dominate on both Cu(111) and Cu(997) now shown in Figure S11. This could include oxygenated hydrocarbons, confusing the interpretation of the O1s, and also the strong overlap of adventitious carbon likely influences the attempts at corroborating CO adsorption by comparing the C1s and O1s intensities. In fact, I was quite surprised by the authors response to reviewer point 3-1: "In this quantitative analysis, we obtained a reliable value of 1.1 ± 0.1 for the atomic ratio of O/C, which confirms the peak evolutions by molecular interactions of CO adsorbates on the Cu(997) surface." I can't see that this can be reliable with the extent of overlap with the adventitious carbon, and this issue didn't seem to be mentioned.*

Response: We appreciate the reviewer's extensive and scholarly opinions for improving our manuscript. We welcome any scientific discussions on unexpected contamination issues in our present study, and we believe our point-by-point responses to the reviewer's arguments have been provided in a scientifically fair and objective manner. We are sorry to hear that our experimental evidence with more added/revised discussions in the previous round could still not fulfill the reviewer's requests on our revised manuscript.

As pointed out by the reviewer, we detected carbon species in C 1s core-level spectra (Supplementary Fig. 11) under 0.2 mbar CO(g) conditions. The labeled C* species at 284.1–284.4 eV are commonly characterized from the Cu(111) and Cu(997) surfaces. Although we minimized

carbon contamination sources from the chamber wall and connected gas lines, the contamination species cannot be fully excluded at ambient pressures. It is well known in *in situ/operando* surface science communities; for instance, the partial amount of hydrocarbons and water molecules in the reaction gas may produce $\sim 10^{-6}$ Torr partial pressures when the total reaction pressures reach a few Torr range in the analysis chamber [G. Ketteler et al., *J. Phys. Chem. C* **111**, 8278-8282 (2007); <https://doi.org/10.1021/jp068606i>].

However, the detected C* species mostly originated from intrinsic carbon impurities of the Cu crystal or dissociated carbon from CO at defect sites, not from oxygenated carbon species. We would like to emphasize that the pronounced peak intensity of C* species from the Cu(997) surface is stronger than that of Cu(111), as shown in Supplementary Fig. 11. In fact, Nilsson and colleagues showed similar results of graphitic carbon evolutions due to the CO dissociation over the stepped Cu surface [M.L. Ng et al., *Phys. Rev. Lett.* **114**, 246101 (2015); <https://doi.org/10.1103/PhysRevLett.114.246101>]. We mentioned this point, “According to previous literature, the appeared peak or spectral shoulder at 529.5–529.6 eV would be assigned to atomic oxygen by dissociation of CO or H₂O. It is reasonable that the introduced CO molecules may have interacted with a limited number of defect sites at the early stage of the overall chemical reaction over the vicinal Cu facets,” in our original manuscript.

Even though the weak peak of labeled C* species is also found in our additional AP-XPS experiments of CO₂/Pt(997), we could not find strong evidence for surface catalysis caused by the carbon species. The deconvoluted C* peak in the C 1s core-level spectrum under 0.8 mbar CO₂(g) has no significant spectral change compared to before and after CO₂(g) interactions. Moreover, the detected peaks of CO* and CO₃* species disappear from the same spectrum because they are not involved in the secondary catalytic reactions, such as water or oxygenated carbon contaminations, as raised the issue by the reviewer.

The reviewer also claimed that the enhanced L₃-edge intensity in XAS measurements at 0.2 mbar CO(g) was strong evidence for the oxidized Cu surface. Actually, the reviewer’s arguments are correlated with Cu 2p core-level with Auger electron transition analyses. That has been extensively discussed in recent publications dealing with confirming the partial amount of Cu(I) or Cu(II) oxides during the electro-reduction of CO₂ [1] P. Grosse et al., *Angew. Chem. Int. Ed.* **57**, 6192-6197 (2018); <https://doi.org/10.1002/anie.201802083>, 2) D. Gao et al., *Angew. Chem. Int. Ed.* **58**, 17047-17053 (2019); <https://doi.org/10.1002/anie.201910155>, 3) W. Fu et al., *ACS Sustainable Chem. Eng.* **8**, 15223-15229 (2020); <https://doi.org/10.1021/acssuschemeng.0c04873>, 4) T. Möller et al., *Angew. Chem. Int. Ed.* **59**, 17974-17983 (2020); <https://doi.org/10.1002/anie.202007136>,].

Their spectroscopic identifications of oxide species from Cu-based nanomaterials show a range of possibilities for the sensitive oxidation state response at the interface during C_1 reduction reactions. Those fancy works adopting nanomaterials have interesting features in that fundamental principles on the basis of surface science could be extended to real applications for developing industrial processes.

However, we need to take a look at them carefully to interpret adsorbate species on the surface based on the fundamental relation of spin-orbit coupling and traveling photoelectrons. We already showed that the characterized peaks of Cu $2p_{3/2}$ and $2p_{1/2}$ had no shift in the previous round. Although the reviewer mentioned in another comment that the analysis of Cu 2p spectra was not relevant to the comparison results due to the indistinct difference between Cu_2O and Cu, the Cu 2p core-level state, their interpretations are also correlated with other adsorbates or scattered inelastic electron information.

Our Cu 2p core-level AP-XP spectra collected in UHV, 0.2 mbar CO(g), and gas evacuation conditions are almost identical to the typically measured clean Cu 2p core-level spectrum in UHV. Of course, we compared our data with other reference data of metallic Cu, Cu_2O , and CuO from other research groups. However, we could not find clear evidence for Cu oxidation phenomena caused by CO adsorbates. In general, we can track gradual peak shifts with the broadening of Cu $2p_{3/2}$ and the evolution of satellite features depending on the progress of Cu oxidation in surface and subsurface regions [M.C. Biesinger, *Surf. Interface Anal.* **49**, 1325-1334 (2017); <https://doi.org/10.1002/sia.6239>]. But it is also difficult to distinguish a clear peak shift for Cu $2p_{3/2}$ of Cu_2O compared to the metallic Cu, because the reported literature values of the peak position difference are usually less than 0.2 eV (within a standard deviation).

In contrast, the analysis of Auger electron transitions or XAS measurements on Cu oxides can support useful information to figure out the oxidation state of the “layered” Cu oxide surface and near-surface regime, depending on characterization methods, with the employed X-ray irradiation depth at the interface. We can find more XAS data in the same line from another reference [P. Jiang et al., *J. Chem. Phys.* **138**, 024704 (2013); <https://doi.org/10.1063/1.4773583>]. Salmeron and colleagues reported representative Cu, Cu_2O , and CuO features very clearly by employing XPS and XAS at the photoemission endstation at beamline 11.0.2 of the Advanced Light Source (ALS) in Berkeley. While the reviewer claimed that our presented data were related to the formation of Cu_2O in XPS and XAS measurements, we could not find supporting evidence for the reviewer’s arguments from previous literature.

Unlike the reviewer’s arguments, our X-ray absorption spectra collected in UHV, 0.2 mbar CO(g), and gas evacuation conditions are almost identical to the metallic Cu data from other published references. Obviously, our spectroscopic analysis results are not matched

with the reference for Cu₂O species in XAS, and the characterized Cu L₃ and L₂ edges have no meaningful shift. Even though the reviewer persistently claimed the recorded signal intensity change of Cu L₃-edge was strong evidence for the oxidized Cu(I) species, the increased peak intensity (enhanced ~16% compared to the metallic Cu) was attenuated again immediately after gas evacuation.

According to theoretical approaches, the path-length distribution function for traveling photoelectrons between initial and final states could be approximated to the inelastic electron scattering cross-section defined as [N. Pauly et al., *Surf. Sci.* **620**, 17-22 (2014); <https://dx.doi.org/10.1016/j.susc.2013.10.009>]:

$$K_{sc}^{XPS}(E, \hbar\omega, \theta) = \frac{\int_0^\infty dx Q(E, \chi, \theta) K_{sc}^{XPS}(E, \hbar\omega, \chi_0, \theta)}{\int_0^\infty dx Q(E, \chi, \theta)}$$

[χ : the total of all path lengths, $Q(E, \chi, \theta)$: the path-length distribution function, $\hbar\omega$: the dispersion relation]

For the case study of Cu oxides, the parameters of optical properties [D. Tahir and S. Tougaard, *J. Phys.: Condens. Matter* **24**, 175002 (2012); <https://doi.org/10.1088/0953-8984/24/17/175002>] can be used for the expansion of this approximation with the dispersion relation in the Drude–Lindhard type oscillators [1) R.H. Ritchie and A. Howie, *Philos. Mag.* **36**, 463 (1977); <https://doi.org/10.1080/14786437708244948>, 2) N. Pauly et al., *Surf. Interface Anal.* **46**, 283-288 (2014); <http://doi.org/10.1002/sia.5411>].

From the model approximations of the multi-scattered electrons, the satellite features of Cu 2p_{3/2} for Cu, Cu₂O, or CuO could be estimated in the primary function. Particularly, Cu(II) oxide has pronounced shake-up features with two overlapping satellites, unlike the single satellite evolution of Cu or Cu(I) oxide. The kinetic energy differences between satellites and the main Cu 2p_{3/2} peak are 8–10 eV for CuO, and ~14 eV for Cu or Cu₂O. These approximation results explain the relationship between possible charge transfer configurations (electron moves from the initial state $|i\rangle$ to the final state $|f\rangle$) of Cu oxides corresponding to total angular momentum in J – J coupling. Although we know that these results cannot reflect the high level of multiplicity, the approximation calculations indicate that the observed satellite features in our Cu 2p core-level AP-XP spectrum ($P_{CO} = 0.2$ mbar at 300 K) are not confined only to the shake-up evolutions caused by Cu oxidation; because the charge transfer configurations for the metallic Cu may also contribute to the formation of satellite features, similar to the case of Cu₂O.

Above all, the observed satellite features in the Cu 2p core-level AP-XP spectrum under 0.2 mbar CO(g) disappear after gas evacuation, which supports that this phenomenon

is related to the surface-electron-gas interaction caused by inelastic scattering of photoelectrons at ambient pressures [A. Jürgensen et al., *J. Electron Spectros. Relat. Phenomena* 232, 111-120 (2019); <https://doi.org/10.1016/j.elspec.2018.11.004>]. The correction algorithms which describe the inelastic scattering of photoelectrons are under development by theory groups, but there are still technical limitations, due to variable inelastic scattering changes as a function of the kinetic energy distribution of the outgoing electrons between sample specimen and cone-aperture of hemispherical electron analyzer [L. Pielsticker et al., *Surf. Interface Anal.* 53, 605-617 (2021); <https://doi.org/10.1002/sia.6947>]. We can look up another case of the surface-electron-gas interaction for 3 mbar CO(g)/Cu(111) from a reference [X. Zhang and S. Ptasinska, *ChemCatChem* 8, 1632-1635 (2016); <https://doi.org/10.1002/cctc.201600046>].

In summary, we have made changes to our manuscript and supplementary information, as follows. The revised paragraphs are also highlighted in the revised manuscript.

Supplementary Fig. 13:

Supplementary Fig. 13 | Synchrotron-based X-ray photoelectron spectroscopy measurements for Cu 2p core-level states (left) and Cu $L_{2,3}$ absorption edge (right) on the Cu(997) surface. The escaped photoelectron and absorbed photon signal intensities were

collected in analysis conditions of UHV, 0.2 mbar CO(g), and after gas evacuation ($h\nu = 1050$ eV; $T = 300$ K). The plotted XPS results indicate no significant change in the oxidation state of Cu 2p_{3/2} and Cu 2p_{1/2}. The spin-orbit couplings difference between Cu 2p_{3/2} and Cu 2p_{1/2} at 933.0 and 952.8 eV ($\Delta = 19.8$ eV) was also kept under 0.2 mbar CO(g) conditions. We observed an unusual species at ~946.0 eV during AP-XPS measurements under 0.2 mbar CO(g), but the satellite features disappeared immediately in the Cu 2p core-level spectrum after gas evacuation. In plotted XAS results, the characterized Cu absorption edges show almost identical spectral shapes under different measurement conditions. The relative intensity changes in the L₃ edge of X-ray absorption spectra under 0.2 mbar CO(g) and gas evacuation conditions correspond to ~1.16 ($I_{\text{red}}/I_{\text{black}}$) and ~1.00 ($I_{\text{blue}}/I_{\text{black}}$), respectively.

Page 16, Line 23:

The relative portion of sp^2 (C=C) or sp^3 (C-C) carbon species in the C 1s core-level spectrum was not considered for this analysis to focus on the detected CO-Cu features in the fingerprint regions of –CO functional groups.

Page 17, Line 7:

Further investigation in X-ray photoelectron spectra for Cu 2p core-level state and X-ray absorption spectra (XAS) on Cu L_{2,3} absorption-edge indicates that the characterized metallic Cu surface (Cu⁰) does not make a transition to layered oxides consisting of Cu₂O (Cu⁺) or CuO (Cu²⁺) with prolonged CO(g) exposure (Supplementary Fig. 13).

Page 17, Line 14:

Besides, the obtained XAS in UHV, 0.2 mbar CO(g), and after gas evacuation conditions do not match with referenced data of Cu oxides. Even though we measured a minor change of signal intensities at the L₃-edge under 0.2 mbar CO(g), the enhanced peak intensity was attenuated again, the same as the original species after gas evacuation. In general, the accepted probing depth by XAS is about 2 nm in Auger electron yield (AEY) mode. In addition, the evidence of Cu oxides can be easily found in regions of the Cu surface and sub-surface because of the thermodynamically favored penetration of adsorbed oxygen. With the nature of Cu-O and the basis of the AEY-XAS principle, our characterization results exhibit no oxide layers in the near-surface layers of the vicinal Cu crystal.

Reviewer comment #2-1: *Whilst I think APXPS studies with CO₂ exposure would strengthen the paper (as does reviewer 3), I accept the authors' arguments about the reversibility of the clustering induced by CO being observed by STM.*

Response #2-1: We thank the reviewer for their constructive suggestions for our manuscript. In this round, we added AP-XPS measurement results of CO₂(g)/Cu(997) to the revised manuscript. **We observed additional peak evolutions of CO* and CO₂^{δ-} in O 1s and C 1s core-level spectra under 0.8 mbar CO₂(g) conditions.** The corresponding morphology structures obtained from AP-STM images aid in understanding adsorbate interactions followed by CO₂ dissociation at Cu step-edge sites. The newly added Supplementary Figure and revised paragraphs in the manuscript are as follows.

Supplementary Fig. 16 | Comparison observation results of CO₂ dissociation over vicinal Cu surfaces probed with AP-XPS and AP-STM. Collected Cu 2p, O 1s, and C 1s core-level AP-XP spectra in UHV (bottom; black line), 0.8 mbar CO₂(g) (middle; red line), and gas evacuation (top; blue line) conditions at 300 K ($h\nu = 1486.7$ eV). (**inset**) Corresponding morphology structures of the Cu(997) surface in UHV, 1 mbar CO₂(g), and gas evacuation conditions. The displayed in situ AP-STM images were sequentially obtained in the same manner as **Figs. 1c-f** at 300 K.

Page 18, Line 21:

Catalytic behaviors of dissociated O and CO adsorbates from CO₂. As shown in previous sections, the surrounding gaseous environmental conditions of O₂ or CO molecules affect the irreversible surface reconstruction or reversible restructuring phenomena on the vicinal Cu surface. In principle, the dissociative adsorption of atomic oxygen and CO adsorbates from CO₂ could be involved in the observed geometry alterations at the Cu step-edge sites (See Supplementary Text). Obtained AP-STM images and AP-XP spectra under the different gaseous conditions could provide consistent evidence for the complementary relations of adsorbed atomic oxygen and CO adsorbates on the Cu(997) surface, and they suggested atomic-scale insights regarding the unexpected surface restructuring under ambient pressure environments. Theoretically, it is predicted that the catalytic reaction pathway of CO₂ conversion is very complicated on Cu-based catalysts, and the dissociation yield of CO₂ could be influenced depending on surface geometries at the early stage of overall catalytic reactions. The catalytic dissociation of CO₂ can simultaneously produce atomic oxygen and CO from the Cu step-edge sites. However, fundamental knowledge of adsorbate molecule interactions at the gas-solid interface has been mainly discussed in metallic or O-covered Cu surfaces with low-index facets in previous literature. The extension of principles only from the low-index facets such as {111}, {110}, and {100} may have a restricted point of view to represent the formation of active sites on rational nanocatalyst designs consisting of vicinal surface geometries and ligand interactions during CO₂ activation processes.

Our spectroscopic and microscopic analysis results reveal the critical role of adsorbate interactions with CO₂ dissociation at under-coordinated Cu sites, as shown in comparison results probed with AP-XPS and AP-STM (Supplementary Fig. 16). Obviously, we find different trends of the Cu clustering under CO₂(g) conditions, compared to the observed Cu(997) surface exposed to O₂(g) or CO(g), as shown in **Figs. 2 and 3**; i.e., a mixed fashion of morphologies between oxidized Cu step-edges (**Fig. 2a**)

and CO-driven Cu nanocluster formations (**Fig. 3c**) on vicinal Cu surfaces. The dissociated O* from CO₂ at Cu step-edge sites may cause irreversible surface roughening due to the formation of atomic-scale CuO_x clusters. Consequently, the following chemical reaction between the adsorbed atomic O* and CO₂ molecule may evolve CO₃-related chemical species at the Cu step-edge sites, affecting local surface restructuring during the catalytic CO₂ activation at ambient pressures. These atomic-scale behaviors under CO₂(g) conditions differ from the reported CO-driven surface restructuring phenomena caused by repulsive CO–CO interactions, as the dissociated O* or CO₃* species also participate in the surface geometry alterations at the step-edge sites of vicinal Cu surfaces.

Supplementary Text

Catalytic behaviors of dissociated O and CO adsorbates from CO₂. Comparison results probed with AP-XPS and AP-STM exhibit the observed catalytic CO₂ activation over vicinal Cu surfaces in order to elucidate CO₂ dissociation and further interactions with dissociated oxygen and CO adsorbates at Cu step-edge sites under gaseous CO₂ environments. In Supplementary Fig. 16, the plotted photoelectron signals in Cu 2p, O 1s, and C 1s core-level spectra (bottom; black line) were collected from the as-cleaned Cu(997) surface in UHV. They are consistent with the synchrotron-based AP-XPS results of the clean vicinal Cu surfaces. Although we characterized a small adventitious carbon feature at 284.5 eV after repeating surface cleaning cycles in UHV, we could not find any critical relationship between the adsorbed carbon species and dissociated adsorbates during the AP-XPS measurements under 0.8 mbar CO₂(g) conditions.

Unlike AP-XPS characterizations on Cu surfaces under CO(g) conditions, we find versatile adsorbate species in O 1s and C 1s core-level AP-XP spectra under the 0.8 mbar CO₂(g) conditions (middle; red line). In addition, the representative peak of gas-phase CO₂ clearly shows up in both spectra. The deconvoluted peaks at 529.9 and 531.1 eV could be attributed to adsorbed atomic O and O-C-O*/C-O* related species. Their correlated characterization results in the C 1s core-level AP-XP spectrum support that the resolved peaks at 286.2 and 288.8 eV are close to the fingerprint regions of Cu-CO and carbonate, formate, or carboxylate species (CO₂^{δ-}) in AP-XPS analyses, respectively. Still, the portion of labeled C* species in the C 1s peak deconvolution has negligible changes compared to the same feature measured in UHV.

After gas evacuation, the evolved features at 529.6–531.4 eV remain in the O 1s core-level spectrum, which would be associated with adsorbed oxygen on the Cu(997) surface. Even though the characterized peak region may overlap with the well-established

hydroxyl group-containing species at 530.8–531.3 eV, the water contamination was not identified in the same spectrum. In addition, the characterized Cu 2p core-level AP-XP spectrum verifies no significant change in Cu 2p_{3/2} and Cu 2p_{1/2} peaks before and after introducing CO₂ gas molecules. We note that the transient evolution of satellite features at 943–948 eV in the Cu 2p core-level AP-XP spectrum is only found under 0.8 mbar CO₂(g) conditions. These unusual features in our present study are irrelevant to the representative shake-up peaks of Cu(I) and Cu(II) oxides. They clearly disappear from the Cu 2p spectrum after gas evacuation and from the synchrotron-based AP-XPS results under CO(g) conditions.

From the collected AP-XP spectra and corresponding morphology images under similar experimental conditions, we could propose the role of dissociated oxygen and CO adsorbates from CO₂ gas molecules on the vicinal Cu surface. Although the CO₂ molecule has a very low chance of effective collisions at the gas-solid interface, the stepped Cu edges can offer active sites for CO₂ activation at ambient pressures. As a result, the dissociated O* and CO* would have an interplay between their adsorption sites and the formation of Cu nanoclusters along the stepped Cu geometry. The dissociated CO* adsorbates may induce step-broken Cu nanoclusters at Cu step-edge sites; in turn, they would evolve reversible surface restructuring at ambient pressures. In contrast, once under-coordinated Cu atoms of the Cu step-edge sites are anchored by dissociated O* from CO₂, the specific sites will be involved in irreversible surface reconstructions. The terminated oxygen of Cu step-edge sites may also react with neighboring CO adsorbates or gaseous CO₂ molecules under ambient CO₂(g) conditions, and the produced carbonate species (–CO₃) could participate in the surface morphology alterations.

Page 23, Line 12:

Laboratory-based AP-XPS measurements. The home-built AP-XPS system with a commercially available X-ray source (a monochromated Al K α anode; $h\nu = 1486.7$ eV) was utilized for investigations of Cu 2p, O 1s, and C 1s core-levels changes on the Cu(997) surface under 0.8 mbar CO₂(g) conditions. The GIST-ESCA group managed and conducted the laboratory-based AP-XPS setup (GIST, the Republic of Korea). The adjusted incident angle of the X-ray beam was 62.5° against the mounted Cu(997) crystal, and the generated photoelectrons were captured at a normal angle (90°) by the 2-D detector of hemispherical electron analyzer (R4000 HiPP-3 APPES analyzer, Scienta Omicron). The clean Cu(997) surface was prepared by the same protocol with AP-STM and synchrotron-based AP-XPS measurements. The sample temperature was monitored using

a directly connected K-type thermocouple attached to the edge of the polished crystal side. A home-built water trap further purified the high-purity CO₂ gas (99.999%) to prevent surface contamination by water impurities. All AP-XP spectra were collected after the AP-XPS system bakeout at 393 K for 120 hrs. Details on the setup and its manipulation procedures were described in the previous work.

Reviewer comment #2-2: *The authors have made some alterations to the manuscript to acknowledge atomic O and hydroxyl group contamination, but they still describe these as small in their response to comment #2-2 and the changes made to the manuscript. However the spectrum the present in Figure 4a of the manuscript has very similar features and intensity ratios to experiments with water adsorption shown in ref 28 at 0.05 torr (except for the H₂O gas phase species isn't present) and included here as a figure for convenience. Given the extent of contamination noted in #2-2 above, and the surface oxidation of Cu noted in #2-4 below, this all points to significant surface contamination when the CO is introduced.*

Response #2-2: We understand the reviewer's concerns on contamination issues. As we mentioned in our response to the reviewer's general comments above, the photoelectron cross-sections can be mainly determined by charge transitions between the initial and final states of surface electronic structures. In principle, those processes are very complicated, depending on the electronic configuration of core-level structures and the material's dielectric and optical properties. Also, the kinetic energy of the X-ray source can affect the charge transfer process during spectroscopic measurements. So, several spectral parameters; background, peak intensity, and peak broadness could be manipulated by various circumstances. They are also influenced by a confined distance between the cone-aperture and specimen surface because the synchrotron-based AP-XPS/XAS system employs the differential pumping stages-equipped hemispherical electron analyzer. **Accordingly, further arguments on the basis of spectral similarity and its intensity ratio in the O 1s core-level interpretations may be uncertain in critical comparisons between our present study and the literature, even if the reviewer suspected the presence of water contamination species from our AP-XP spectra.**

In addition, we also showed the control experiment results of Cu(111) under 0.2 mbar CO(g) in our manuscript to compare unique properties of the CO-induced surface restructuring on the Cu(997) surface. We conducted AP-XPS measurements of CO(g)/Cu(111) in the same procedure after collecting AP-XP spectra from the Cu(997) surface at the same beamline endstation. **However, we could not detect any similar spectral features of the "active" CO/Cu(111) surface with 0.05 Torr H₂O(g) in ref. 28, because we did not introduce water molecules into**

the analysis chamber. Again, we already have shown the obtained data of CO(g)/Cu(997) and CO(g)/Cu(111) in our original and revised manuscript to appeal this aspect, considering the suspected contamination issues in surface science communities.

We think the contamination-related arguments raised by the reviewer could have originated from radical issues of the O 1s core-level structure analysis in XPS measurements [H. Idriss, *Surf. Sci.* **712**, 121894 (2021); <https://doi.org/10.1016/j.susc.2021.121894>]. We can search for similar long-standing problems in previous literature concerning environmental contaminations probed with AP-XPS [L. Trotochaud et al., *J. Phys. Chem. B* **122**, 1000-1008 (2018); <https://doi.org/10.1021/acs.jpcc.7b10732>] or X-ray beam radiation damage effects on the oxidized Cu surface at the liquid-solid interface [R.S. Weatherup et al., *J. Phys. Chem. B* **122**, 737-744 (2018); <https://doi.org/10.1021/acs.jpcc.7b06397>].

If a significant amount of water contamination influenced our AP-XPS data of CO(g)/Cu surfaces, the AP-XP spectra from measurement results of CO(g)/Cu(111) should also be almost identical to one of the spectra for the “activated” surface from ref. 28. However, we could not identify any water-induced activation phenomenon from our CO(g)/Cu(111) data. Although the evolved peak features at 531.2 eV overlap with the literature values of the hydroxyl group, we minimized water contamination issues before we conducted AP-XPS measurements, with multiple bakeout procedures for all chambers and connected gas lines. We understand that a small amount of water contamination from somewhere in the dead volume of chambers or gas line connections could possibly arise, even if we had sufficiently baked all systems. Thus, to accommodate the reviewer’s comments, our manuscript already discussed possible minor issues caused by water contamination.

Reviewer comment #2-3: *The authors have not fully addressed the issue I was trying to refer to, they discuss the Cu2p spectra, but it is well established that Cu⁰ and Cu⁺ are not readily distinguished by measurements of the Cu2p regions, so comparisons of this are not really relevant. I was instead referring to the NEXAFS, and therefore realize my comment may have been ambiguous (including some small typographical errors). I therefore restate my point here more clearly:*

The Cu L-edge NEXAFS in Figure S13 (right hand panel) in the revised manuscript shows a clear change between UHV and 0.2 mbar CO which is consistent with a transition toward Cu₂O. In fact the change is much greater than seen in the Supplementary Information of ref 16 during CO₂ exposure at a similar pressure, where O is expected to be supplied by CO₂ dissociation. Below is a comparison of this, where I have shifted one of the authors spectra so they are overlaid for more clear comparison (left hand panel), and included the data from ref 16 (right hand panel).

Based on a rough measurement of the spectra in the current work, the intensity difference between the first peak and trough is ~1.4x larger when the CO is added compared to under UHV. For reference 16, the increase is only 1.05x following CO₂ exposure. Both measurements are AEY-NEXAFS so arguments about different depth sensitivities are not relevant. It should also be noted that the L₃ edge position is also shifted to slightly higher energy after the CO exposure in the current work. This indicates oxidation of the Cu surface to form Cu₂O, however the authors do not account for this and suggest that they “cannot imagine any other oxygen source except for the negligible amount of atomic O from the CO dissociation.”

Response #2-3: We appreciate the reviewer's careful interpretations of our XAS data and deep interest in improving our manuscript. Before we discuss more details regarding the reviewer's comments, we would like to define the meaning of copper oxides in our present study. Generally, the Cu₂O is close to such a layered oxide structure, as shown in previous literature [1] F. Yang et al., *J. Phys. Chem. C* **114**, 17042-17050 (2010); <https://doi.org/10.1021/jp1029079>, 2) A.J. Therrien et al., *J. Phys. Chem. C* **120**, 10879-10886 (2016); <https://doi.org/10.1021/acs.jpcc.6b01284>]. Because we only observed the submonolayer of adsorbed CO or O* on the Cu(997) surface in AP-STM and AP-XPS measurements, we preferred the description of adsorbed O* on the Cu surface rather than the label of Cu₂O for such small oxidic Cu species. However, a reference that dealt with the initial oxidation of vicinal Cu surfaces using a “curved” Cu crystal also described the growth of the Cu₂O step oxide between 1–2 nm boundaries of the Cu surface [T.J. Lawton et al., *J. Phys. Chem. C* **116**, 16054-16062 (2012); <https://doi.org/10.1021/jp303488f>]. So, we used the term of “CuO_x” or “oxide-like Cu₂O layer” in our manuscript to discuss the initial oxidation of Cu(997) under O₂(g) environments.

In fact, we mainly mentioned the concept of “adsorbed O*” on the Cu(997) under O₂(g) or CO₂(g) conditions. We did not discuss the oxidation of Cu(997) under CO(g) conditions, because we focused on the reversible surface reconstruction phenomenon by CO–CO interactions at ambient pressures. Even though the dissociation of CO at the Cu step-edge could provide dissociated O* from CO, the total amount of adsorbed oxygen would be less than 0.1 ML on the Cu(997) surface. **As we responded to the reviewer's general comments above, the changes in signal intensities in our XAS data cannot be explicated with strong evidence of Cu oxidation because the enhanced signal intensity was attenuated to the measured level in UHV again after gas evacuation.** Thus, we already explained that the instant signal intensity change originated from the inelastic photoelectron scattering at the gas-solid interface at ambient pressures, not caused by Cu oxidation.

We carefully considered the supporting information from ref. 16 (the described inelastic mean-free path by the authors is ~ 1.5 nm) that the presented NEXAFS Cu L_3 edge of bare Cu(100), Cu₂O, and Cu(111) surface under 0.3 Torr CO₂(g). The authors mentioned such sensitive peak intensity changes in the main edge and resonance peaks. **As mentioned by the authors from ref. 16, there is no discernable difference between their NEXAFS spectra of the Cu L_3 -edge. The authors claimed such “a very slight oxidation to Cu₂O” at the edge of Cu nanoclusters in the ref. 16, and they described them as “some parts of the surface being oxidized to Cu₂O” in main contexts.** Even though their collected signal intensity changes in NEXAFS spectra are very small, the attenuated resonance peak trend looks somewhat similar to the oxidation of the Cu surface in another reference [S. Kunze et al., *Chem. Sci.* **12**, 14241-14253 (2021); <https://doi.org/10.1039/d1sc04861a>]. Based on this rationale, the authors from ref. 16 could have extended their arguments regarding the oxidation of Cu(100) by dissociated oxygen from CO₂(g) at ambient pressures.

However, we cannot find analogous resonance peak changes in our Cu L_3 -edge measurements. In fact, the recorded signal intensity of the Cu L_3 -edge was only enhanced at 0.2 mbar CO(g), but we could not detect any discernible peak shift or changes in other resonance peaks. We find a similar trend of Cu(997) under 0.8 mbar CO₂(g) in additional AP-XPS results.

Reviewer comment #2-4: I agree that for a clean chamber and CO gas line, Cu oxidation is not expected given CO is highly reducing and should in fact remove oxygen from the Cu surface. Instead, the fact they do see oxidation suggests a significant source of contamination, likely water or oxygenated hydrocarbons being displaced from the walls of the chamber/gas line. This is further supported by the significant adventitious carbon they see appear in the C1s region. This calls into question the much of the APXPS results within this manuscript, as H₂O, CH_x, and CH_xO_y are likely to overlap with many of the features being attributed.

Response #2-4: As discussed in our response to the general comment, we could not characterize any significant oxidation process on the Cu(997) surface under 0.2 mbar CO(g) environments. We have already provided additional experimental data and discussions, and the possible water contamination issues were mentioned in the previous round. We also know that the contamination issue regarding oxygenated hydrocarbons should be carefully considered to interpret AP-XPS measurements during catalytic C₁ reactions. Recent literature on the methanol conversion topic explains the critical case regarding the variable adsorbate coverages caused by the oxygenated hydrocarbon contaminations on the Cu surface [B. Eren et al., *PCCP* **22**, 18806-18814

(2020); <https://doi.org/10.1039/d0cp00347f>]. Moreover, as mentioned by the reviewer, accumulated water contamination on Cu oxides may affect the chemical process as a function of relative humidities. In this case, the contaminated oxide surfaces probably readily react with the adventitious carbon even at low relative humidity [L. Trotochaud et al., *J. Phys. Chem. B* **122**, 1000-1008 (2018); <https://doi.org/10.1021/acs.jpcc.7b10732>]. However, our conducted experimental conditions are far from those humid oxidation environments.

Response to Reviewer #3:

General comment: *I appreciate the authors effort to reply to my comments and revise the manuscript. I am satisfied with the responses #3-3, 3-4, 3-5, but not with the responses #3-1, 3-2. Especially, in the response #3-1, the authors chose not to show O 1s and C 1s AP-XPS spectra of Cu(997) in 1 mbar CO₂ in the revised manuscript, which were important to support their conclusion and prove the novelty of the work as discussed below. Therefore, I could not recommend its publication in Nature Communications in the current form.*

Response: We thank the reviewer for their constructive criticism of our manuscript. We understand the reviewer's concerns regarding the novelty of our manuscript and conclusive proof of the interplay between dissociated adsorbates from CO₂ and surface morphologies. In the previous round, we added supplementary data of C 1s core-level AP-XP spectra with their analysis results to prove CO-induced Cu nanocluster evolutions on the Cu(997) surface, but we did not support additional AP-XPS data under CO₂(g) environments.

To our best knowledge, the previous literature on CO₂/Cu(997) already proved the CO₂ dissociation over the vicinal Cu surface using AP-XPS [T. Koitaya et al., *Top. Catal.* **59**, 526-531 (2016); <https://doi.org/10.1007/s11244-015-0535-1>]. In addition, another comparison result of AP-XPS measurements on Cu(100) and Cu(111) surfaces under CO₂(g) conditions also showed superior properties of the stepped Cu surface compared to flat terraces [B. Eren et al., *JACS* **138**, 8207-8211 (2016); <https://doi.org/10.1021/jacs.6b04039>]. Based on these studies, we assumed that we did not need further AP-XPS investigation on CO₂(g)/Cu(997), and we focused on extensive AP-XPS analyses of CO(g)/Cu(997).

We agree with the reviewer's request for additional AP-XPS experiments under CO₂(g) conditions. It is reasonable to prove our claims on surface catalysis of CO₂/Cu(997), and we need our own AP-XPS data in order to propose a critical role of surface morphology and catalytic interplays between chemical species and vicinal Cu geometries. **Thus, we added more AP-XPS data of CO₂/Cu(997) to our revised manuscript, as requested by the reviewer.** For this reason, we have made major changes in sections related to AP-XPS analysis to reflect supplementary data supporting our scientific rationale. Please see the following point-by-point response and discussion.

Reviewer comment #3-1: *The authors concluded that the surface morphology change of Cu(997) in near-ambient CO₂ gas originates mainly from CO adsorbates, forming Cu_x(CO)_y clusters; atomic O atoms also have small contributions to it. This conclusion is based on solely on AP-STM topographic images that show the reversible and irreversible behaviors of the Cu(997) surface in*

CO and O₂, respectively. I think that this is not strong enough to support their conclusion and additional spectroscopic evidence by O 1s and C 1s AP-XPS spectra of Cu(997) in 1 mbar CO₂ is necessary.

Response #3-1: As mentioned by the reviewer, our AP-STM images clearly show irreversible surface reconstruction and reversible clustering depending on surrounding gas conditions at ambient pressures. The adsorbed atomic oxygen at Cu step-edge sites hinders further interactions with surrounding gas molecules, whereas the elastic interactions between CO adsorbates at Cu step-edges cause reversible surface restructuring as a function of CO(g) pressures. Even though the adsorbate-driven morphologic alterations were solely confirmed under O₂(g) or CO(g) conditions, those surface catalysis originating from CO₂ dissociation would happen in higher complexities at the molecular level.

So, we carried out additional AP-XPS experiments of CO₂(g)/Cu(997) to understand the catalytic interactions of dissociated O* and CO adsorbates at Cu step-edge sites. The additionally collected Cu 2p, O 1s, and C 1s core-level AP-XP spectra provide spectroscopic evidence of CO-Cu and carbonate (CO₃) species under 0.8 mbar CO₂(g) conditions at 300 K.

Remarkably, the dissociated O* species remains on the Cu(997) surface after gas evacuation, which is consistent with our rationale based on AP-STM observations at 1 mbar CO₂(g). As shown in AP-STM images, the dissociated O* and CO adsorbates may simultaneously induce surface roughening Cu(997). Further investigated AP-XPS results support the observed vicinal Cu structures under CO₂(g) conditions, as below.

Supplementary Fig. 16 | Comparison observation results of CO_2 dissociation over vicinal Cu surfaces probed with AP-XPS and AP-STM. Collected Cu 2p, O 1s, and C 1s core-level AP-XP spectra in UHV (bottom; black line), 0.8 mbar $\text{CO}_2(\text{g})$ (middle; red line), and gas evacuation (top; blue line) conditions at 300 K ($h\nu = 1486.7$ eV). (**inset**) Corresponding morphology structures of the Cu(997) surface in UHV, 1 mbar $\text{CO}_2(\text{g})$, and gas evacuation conditions. The displayed *in situ* AP-STM images were sequentially obtained in the same manner as **Figs. 1c-f** at 300 K.

Reviewer comment #3-2: *In addition, the chemical nature of the reconstructed Cu(997) surface is very important to prove the novelty of the present study. In my opinion, the formation of metal nanoclusters, step breaking, and its reversible behavior by one type of adsorbates have been all reported in the literature (e.g., $\text{CO}/\text{Pt}(557)$, *Science* 327, 850 (2010).) and lack in novelty. The novelty of this work could be that the surface morphology change is driven by two or more types*

of adsorbates. Therefore, it is of particular important to know how different surface adsorbates of CO and O (and possibly CO₃) compete or cooperate to reconstruct the surface using AP-XPS.

Response #3-2: As pointed out by the reviewer, the CO-induced surface restructuring phenomenon was reported in 2010 [F. Tao et al., *Science* **327**, 850-853 (2010); <https://doi.org/10.1126/science.118212>]. Somorjai and colleagues proposed the CO pressure-dependent behavior correlated with reversible surface clustering caused by repulsive CO–CO interactions. Their spectroscopic analysis results combined with AP-STM observations from UHV to ambient pressures were reproduced again in 2016 by Park and colleagues [J. Kim et al., *JACS* **138**, 1110-1113 (2016); <https://dx.doi.org/10.1021/jacs.5b10628>].

In fact, their physical meaning with repulsive CO–CO interactions on the Cu(997) surface is totally different from the Pt(557) surface. Of course, the Cu surface cannot have high CO coverages as much as CO/Pt surfaces at the same CO(g) pressure at 300 K because the back-donation of π -electrons is not much preferred on Cu surfaces. Instead, the relatively weaker binding energy of Cu-Cu compared to Pt-Pt enables Cu surface restructuring even at a low CO coverage. Salmeron and colleagues revealed this scientific point on the Cu(111) surface at ambient pressures [B. Eren et al., *Science* **351**, 475-478 (2016); <https://dx.doi.org/10.1126/science.aad8868>]. In this regard, the reviewer's comment regarding the novelty is reasonable because the commonly observed CO-driven surface restructuring phenomena seem like ordinary properties.

In contrast, we show the more advanced properties of surface clustering that dissociated O* and CO adsorbates from CO₂ molecules simultaneously contribute to the complicated surface roughening of Cu(997) under CO₂(g) conditions. We find the unique surface restructuring of vicinal Cu(997) driven by two different adsorbates. We also discuss the role of carbonate (CO₃) on vicinal Cu surfaces in the aspect of surface restructuring. To aid a comprehensive understanding of chemical species, we added AP-XPS results of CO₂/Cu(997) to the revised manuscript, as mentioned in **Response #3-1**. The collected Cu 2p, O 1s, and C 1s core-level spectra are displayed with the corresponding surface geometry at each measurement condition of UHV, CO₂(g), or gas evacuation. Please see our additional results and discussions on AP-XPS measurements of CO₂(g)/Cu(997) as follows.

Page 18, Line 21:

Catalytic behaviors of dissociated O and CO adsorbates from CO₂. As shown in previous sections, the surrounding gaseous environmental conditions of O₂ or CO molecules affect the irreversible surface reconstruction or reversible restructuring phenomena on the vicinal Cu surface. In principle, the dissociative adsorption of atomic

oxygen and CO adsorbates from CO₂ could be involved in the observed geometry alterations at the Cu step-edge sites (See Supplementary Text). Obtained AP-STM images and AP-XP spectra under the different gaseous conditions could provide consistent evidence for the complementary relations of adsorbed atomic oxygen and CO adsorbates on the Cu(997) surface, and they suggested atomic-scale insights regarding the unexpected surface restructuring under ambient pressure environments. Theoretically, it is predicted that the catalytic reaction pathway of CO₂ conversion is very complicated on Cu-based catalysts, and the dissociation yield of CO₂ could be influenced depending on surface geometries at the early stage of overall catalytic reactions. The catalytic dissociation of CO₂ can simultaneously produce atomic oxygen and CO from the Cu step-edge sites. However, fundamental knowledge of adsorbate molecule interactions at the gas-solid interface has been mainly discussed in metallic or O-covered Cu surfaces with low-index facets in previous literature. The extension of principles only from the low-index facets such as {111}, {110}, and {100} may have a restricted point of view to represent the formation of active sites on rational nanocatalyst designs consisting of vicinal surface geometries and ligand interactions during CO₂ activation processes.

Our spectroscopic and microscopic analysis results reveal the critical role of adsorbate interactions with CO₂ dissociation at under-coordinated Cu sites, as shown in comparison results probed with AP-XPS and AP-STM (Supplementary Fig. 16). Obviously, we find different trends of the Cu clustering under CO₂(g) conditions compared to the observed Cu(997) surface exposed to O₂(g) or CO(g), as shown in **Figs. 2** and **3**; i.e., there is a mixed fashion of morphologies between oxidized Cu step-edges (**Fig. 2a**) and CO-driven Cu nanocluster formations (**Fig. 3c**) on vicinal Cu surfaces. The dissociated O* from CO₂ at Cu step-edge sites may cause irreversible surface roughening due to the formation of atomic-scale CuO_x clusters. Consequently, the following chemical reaction between the adsorbed atomic O* and CO₂ molecule may evolve CO₃-related chemical species at the Cu step-edge sites, affecting local surface restructuring during the catalytic CO₂ activation at ambient pressures. These atomic-scale behaviors under CO₂(g) conditions differ from the reported CO-driven surface restructuring phenomena caused by repulsive CO–CO interactions, as the dissociated O* or CO₃* species also participate in the surface geometry alterations at the step-edge sites of vicinal Cu surfaces.

Supplementary Text

Catalytic behaviors of dissociated O and CO adsorbates from CO₂. Comparison results probed with AP-XPS and AP-STM exhibit the observed catalytic CO₂ activation

over vicinal Cu surfaces in order to elucidate CO₂ dissociation and further interactions with dissociated oxygen and CO adsorbates at Cu step-edge sites under gaseous CO₂ environments. In Supplementary Fig. 16, the plotted photoelectron signals in Cu 2p, O 1s, and C 1s core-level spectra (bottom; black line) were collected from the as-cleaned Cu(997) surface in UHV. They are consistent with the synchrotron-based AP-XPS results of the clean vicinal Cu surfaces. Although we characterized a small adventitious carbon feature at 284.5 eV after repeating surface cleaning cycles in UHV, we could not find any critical relationship between the adsorbed carbon species and dissociated adsorbates during the AP-XPS measurements under 0.8 mbar CO₂(g) conditions.

Unlike AP-XPS characterizations on Cu surfaces under CO(g) conditions, we find versatile adsorbate species in O 1s and C 1s core-level AP-XP spectra under the 0.8 mbar CO₂(g) conditions (middle; red line). In addition, the representative peak of gas-phase CO₂ clearly shows up in both spectra. The deconvoluted peaks at 529.9 and 531.1 eV could be attributed to adsorbed atomic O and O-C-O*/C-O* related species. Their correlated characterization results in the C 1s core-level AP-XP spectrum support that the resolved peaks at 286.2 and 288.8 eV are close to the fingerprint regions of Cu-CO and carbonate, formate, or carboxylate species (CO₂^{δ-}) in AP-XPS analyses, respectively. Still, the portion of labeled C* species in the C 1s peak deconvolution has negligible changes compared to the same feature measured in UHV.

After gas evacuation, the evolved features at 529.6–531.4 eV remain in the O 1s core-level spectrum, which would be associated with adsorbed oxygen on the Cu(997) surface. Even though the characterized peak region may overlap with the well-established hydroxyl group-containing species at 530.8–531.3 eV, the water contamination was not identified in the same spectrum. In addition, the characterized Cu 2p core-level AP-XP spectrum verifies no significant change in Cu 2p_{3/2} and Cu 2p_{1/2} peaks before and after introducing CO₂ gas molecules. We note that the transient evolution of satellite features at 943–948 eV in the Cu 2p core-level AP-XP spectrum is only found under 0.8 mbar CO₂(g) conditions. These unusual features in our present study are irrelevant to the representative shake-up peaks of Cu(I) and Cu(II) oxides. They disappear very clearly from the Cu 2p spectrum after gas evacuation as well as our synchrotron-based AP-XPS results under CO(g) conditions.

From the collected AP-XP spectra and corresponding morphology images under similar experimental conditions, we could propose the role of dissociated oxygen and CO adsorbates from CO₂ gas molecules on the vicinal Cu surface. Although the CO₂ molecule has a very low chance of effective collisions at the gas-solid interface, the stepped Cu

edges can offer active sites for CO₂ activation at ambient pressures. As a result, the dissociated O* and CO* would have an interplay between their adsorption sites and the formation of Cu nanoclusters along the stepped Cu geometry. The dissociated CO* adsorbates may induce step-broken Cu nanoclusters at Cu step-edge sites; in turn, they would evolve reversible surface restructuring at ambient pressures. In contrast, once under-coordinated Cu atoms of the Cu step-edge sites are anchored by dissociated O* from CO₂, the specific sites will be involved in irreversible surface reconstructions. The terminated oxygen of Cu step-edge sites may also react with neighboring CO adsorbates or gaseous CO₂ molecules under ambient CO₂(g) conditions, and the produced carbonate species (–CO₃) could participate in the surface morphology alterations.

Page 23, Line 12:

Laboratory-based AP-XPS measurements. The home-built AP-XPS system with a commercially available X-ray source (a monochromated Al *ka* anode; $h\nu=1486.7$ eV) was utilized for investigations of Cu 2p, O 1s, and C 1s core-levels changes on the Cu(997) surface under 0.8 mbar CO₂(g) conditions. The GIST-ESCA group managed and conducted the laboratory-based AP-XPS setup (GIST, the Republic of Korea). The adjusted incident angle of the X-ray beam was 62.5° against the mounted Cu(997) crystal, and the generated photoelectrons were captured at a normal angle (90°) by the 2-D detector of hemispherical electron analyzer (R4000 HiPP-3 APPES analyzer, Scienta Omicron). The clean Cu(997) surface was prepared by the same protocol with AP-STM and synchrotron-based AP-XPS measurements. The sample temperature was monitored using a directly connected K-type thermocouple attached to the edge of the polished crystal side. A home-built water trap further purified the high-purity CO₂ gas (99.999%) to prevent surface contamination by water impurities. All AP-XP spectra were collected after the AP-XPS system bakeout at 393 K for 120 hrs. Details on the setup and its manipulation procedures were described in the previous work.

REVIEWER COMMENTS

Reviewer #2 (Remarks to the Author):

The authors further comments and responses have addressed most of my outstanding concerns/questions. Particularly the further APXPS data for CO₂ exposures and the addition of the NEXAFS data following CO removal in Figure S13. I therefore believe the paper to now be suitable for publication in Nature Communications, and congratulate the authors on their manuscript.

The only further optional suggestion I have is that the authors should add a sentence or two explaining the reversible change they are seeing to the Cu L edge when CO is present compared to when it is not. When the CO is present the Cu L edge NEXAFS changes to look more like a partially oxidised Cu surface that could be fitted by a linear combination of the pure Cu and Cu₂O spectra, similar to Figure 1d of DOI: 10.1021/jacs.5b07451. In their responses so far the authors have wanted to avoid relating this to an oxide formation but they haven't really offered an alternative explanation (they focussed on the additional features in the Cu2p XPS, rather than this NEXAFS change). I wonder if this might relate to some electron transfer associated with CO adsorption?

Reviewer #3 (Remarks to the Author):

I would like to thank the authors for their great effort to reply to my request on additional data in the previous round; they performed additional AP-XPS experiments using laboratory X-ray source and showed O 1s, C 1s, Cu 2p AP-XPS spectra of Cu(997) in 0.8 mbar CO₂. The authors claimed that both dissociated O and CO adsorbates from CO₂ molecules contribute to the surface morphology change of Cu(997) under CO₂ gas in a complicated way. Although they did not discuss the detailed interplay between dissociated O and CO adsorbates in the surface morphology change, their arguments are scientifically fair. Therefore, I would recommend its publication in Nature Communications after the following points are addressed by the authors. My comments are limited on the newly-added AP-XPS spectra of CO₂/Cu(997).

1) In the O 1s AP-XPS spectrum of Cu(997) in 0.8 mbar CO₂ (Supplementary Fig. 16), the dominant surface species was atomic O, which leads to irreversible surface reconstruction. However, AP-STM of CO₂/Cu(997) showed reversible surface reconstruction. The authors should give an explanation on this discrepancy.

2) The authors referred to the previous AP-XPS study of CO₂/Cu(997) [Ref. 9 in the main text; T. Koitaya et al., Top. Catal. 59, 526-531 (2016)]. O 1s and C 1s AP-XPS spectra of CO₂/Cu(997) in Ref. 9 seem to be different from Supplementary Fig. 16 in this study; the dominant surface species was CO₃ in Ref. 9. It is

fair to point out the difference.

3) In the caption of Supplementary Fig. 16, please add the lapse time after introducing CO₂. Since there is time-evolution of surface morphology and stoichiometry, the lapse time is necessary to compare AP-XPS spectra with AP-STM data.

Response to Reviewer #2:

General comment: *The authors further comments and responses have addressed most of my outstanding concerns/questions. Particularly the further APXPS data for CO₂ exposures and the addition of the NEXAFS data following CO removal in Figure S13. I therefore believe the paper to now be suitable for publication in Nature Communications, and congratulate the authors on their manuscript.*

Response: We thank Reviewer 2 for the favorable recommendation on our manuscript to be published in *Nature Communications*. Our additional spectroscopic data in Supplementary Fig. 13 should fully address the reviewer's major concerns about analyses of our AP-XPS/XAS measurement results on the vicinal Cu(997) surface.

Reviewer comment #2-1: *The only further optional suggestion I have is that the authors should add a sentence or two explaining the reversible change they are seeing to the Cu L edge when CO is present compared to when it is not. When the CO is present the Cu L edge NEXAFS changes to look more like a partially oxidised Cu surface that could be fitted by a linear combination of the pure Cu and Cu₂O spectra, similar to Figure 1d of DOI: 10.1021/jacs.5b07451. In their responses so far the authors have wanted to avoid relating this to an oxide formation but they haven't really offered an alternative explanation (they focussed on the additional features in the Cu2p XPS, rather than this NEXAFS change). I wonder if this might relate to some electron transfer associated with CO adsorption?*

Response #2-1: We appreciate the reviewer's detailed technical comments on X-ray absorption spectroscopy. We carefully considered the research article suggested by the reviewer [B. Eren et al., *JACS* **137**, 11186-11190 (2015); <https://doi.org/10.1021/jacs.5b07451>] to improve our experimental interpretations in Supplementary Fig. 13. Even though our present study does not directly deal with CO oxidation reaction pathways on the Cu surface, adsorbate CO-driven surface catalysis phenomenon could be associated with our current study due to the unique electron transfers at the gas-solid interface at ambient pressures. Moreover, recently reported theoretical prediction results of CO-driven Cu atoms' migration process implicate that the reactive conditions determine the kinetics of surface catalysis by creating a new type of active site on the surface under relevant pressures and temperatures [L. Xu et al., *Science* **380**, 70-76 (2023);

<https://doi.org/10.1126/science.add0089>]. Based on these points, we added more text to the revised manuscript, as seen below.

Page 17, Line 18:

Technically, the accepted value of probing depth by XAS is approximately 2 nm in Auger electron yield (AEY) mode^{51,52}. In previous literature, we can find a similar feature of the pronounced resonance peak due to the formation of a thin (~1 nm) Cu₂O layer during XAS measurements under CO oxidation conditions [p{O₂} = 0–0.2 mbar; p{CO} = 0.4 mbar]⁵³. The spectroscopic feature of Cu(I) or Cu(II) oxides can be easily distinguished in regions of the Cu surface and sub-surface because of the thermodynamically favored penetration of adsorbed oxygen⁵⁴. With the nature of Cu-O and the basis of the AEY-XAS principle, our XAS results collected before and after introducing 0.2 mbar CO(g) exhibit no significant evidence for the formation of Cu(I) or (II) oxide structures in the near-surface layers of the vicinal Cu crystal. However, the sensitive change of Cu L_{2,3}-edge is only detectable at 0.2 mbar CO(g). A kinetic Monte Carlo simulation study⁵⁵ provides theoretical insight regarding the unusual feature in the first resonance peak of Cu L₃-edge under CO(g) conditions. Because the CO-driven migration of surface Cu atoms at under-coordinated sites cause creating a new type of active site, the enhanced electron transfer property caused by CO adsorbates at 0.2 mbar CO(g) would have been involved in the transient changes in our XAS measurements.

Response to Reviewer #3:

General comment: *I would like to thank the authors for their great effort to reply to my request on additional data in the previous round; they performed additional AP-XPS experiments using laboratory X-ray source and showed O 1s, C 1s, Cu 2p AP-XPS spectra of Cu(997) in 0.8 mbar CO₂. The authors claimed that both dissociated O and CO adsorbates from CO₂ molecules contribute to the surface morphology change of Cu(997) under CO₂ gas in a complicated way. Although they did not discuss the detailed interplay between dissociated O and CO adsorbates in the surface morphology change, their arguments are scientifically fair. Therefore, I would recommend its publication in Nature Communications after the following points are addressed by the authors. My comments are limited on the newly-added AP-XPS spectra of CO₂/Cu(997).*

Response: We appreciate Reviewer 3's recommendation to publish our research article in *Nature Communications*. We believe our additional AP-XPS measurements under 0.8 mbar CO₂(g) are supportive of evidence for the molecular-level interplay between the dissociated CO and atomic oxygen from CO₂ on the Cu(997) surface. As requested by the reviewer, we added more discussions to our revised manuscript and supplementary information to address a couple of minor issues raised by Reviewer 3, as follows.

Reviewer comment #3-1: *1) In the O 1s AP-XPS spectrum of Cu(997) in 0.8 mbar CO₂ (Supplementary Fig. 16), the dominant surface species was atomic O, which leads to irreversible surface reconstruction. However, AP-STM of CO₂/Cu(997) showed reversible surface reconstruction. The authors should give an explanation on this discrepancy.*

Response #3-1: We thank Reviewer 3 for their constructive comments on our manuscript. From the recorded AP-XPS results, the amount of detected atomic oxygen in the O 1s core-level spectrum under 0.8 mbar CO₂(g) seems higher than other species, but the atomic oxygen was still adsorbed on a limited number of under-coordinated Cu atoms after CO₂ dissociation over Cu step-edges at 300 K. Still, almost all the vicinal Cu surfaces had a metallic property during AP-STM observations at 1 mbar CO₂(g) or AP-XPS measurements at 0.8 mbar CO₂(g). In addition, even the dissociated oxygen would also be desorbed to CO₂(g) again due to the spontaneous reaction with a nearby CO adsorbate at under-coordinated Cu atom sites. However, the altered surface morphology structures showed different trends after CO₂(g) or CO(g) evacuation from the reaction cell, as displayed in Supplementary Fig. 4. We could find irregularly broadened step-edge sites (indicated by red arrows

on the AP-STM image) on the Cu(997) surface after CO₂(g) evacuation, whereas the observed Cu(997) surface after CO(g) evacuation was quite organized (indicated by transparent red lines) similar to Cu step-edge sites measured in UHV. These noticeable features could cause a discrepancy regarding irreversible surface restructuring by atomic oxygen and reversible Cu clustering by CO adsorbates from CO₂ gas molecules. We added a sentence to address this point in the revised Supplementary Text, as below.

Supplementary Text – Catalytic behaviors of dissociated O and CO adsorbates from CO₂:

Eventually, we could find the more roughened step-edge geometries after CO₂(g) evacuation, compared to the vicinal Cu surface after CO(g) evacuation, as shown in Supplementary Fig. 4.

Reviewer comment #3-2: *2) The authors referred to the previous AP-XPS study of CO₂/Cu(997) [Ref. 9 in the main text; T. Koitaya et al., Top. Catal. 59, 526-531 (2016)]. O 1s and C 1s AP-XPS spectra of CO₂/Cu(997) in Ref. 9 seem to be different from Supplementary Fig. 16 in this study; the dominant surface species was CO₃ in Ref. 9. It is fair to point out the difference.*

Response #3-2: Our present AP-XPS data collected under 0.8 mbar CO₂(g) conditions shows both detected species of atomic oxygen and CO₃*. However, our spectroscopic probing depth during lab-based AP-XPS measurements was much deeper in comparison with that of Ref. 9. This technical point could cause a different trend in surface characterization results, even if we observed the vicinal Cu surface under almost similar thermodynamic reaction conditions. Regardless of the technical limitation of employing the monochromatic Al K α X-ray source, our experimental analysis results are still supportive of the influence of CO₃* species during surface restructuring phenomena at 300 K, as mentioned in our manuscript. Accordingly, we added more discussions to the revised manuscript in order to elucidate the different spectroscopic trends in characterizing adsorbate species of atomic oxygen and CO₃*, as follows.

Page 20, Line 5:

According to the reported synchrotron-based AP-XPS measurements, the formation of CO₃* species was much more pronounced than atomic oxygen in O 1s core-level AP-XP spectra under 0.8 mbar CO₂(g) at 340 K⁹. In the present AP-XPS study (Supplementary Fig. 16), we also detected both atomic oxygen and CO₃* peaks caused by CO₂ dissociation

at 300 K, but plotted photoelectron signals in the O 1s core-level spectra indicate an opposite trend in the characterization of adsorbate species compared to the previous literature. This would originate from the technical issue for characterizing CO₃* species due to the different probing depths between the synchrotron-based photon source ($h\nu = 630$ eV) and monochromatic Al K α X-ray source ($h\nu = 1486.7$ eV).

Reviewer comment #3-3: *3) In the caption of Supplementary Fig. 16, please add the lapse time after introducing CO₂. Since there is time-evolution of surface morphology and stoichiometry, the lapse time is necessary to compare AP-XPS spectra with AP-STM data.*

Response #3-3: As pointed out by the reviewer, we observed a time-lapse Cu clustering phenomenon on the vicinal Cu surface under 1 mbar CO₂(g) using AP-STM. Because the dissociated CO and atomic oxygen from CO₂ simultaneously react with Cu step-edge sites, we could find partially oxidized features and reversible Cu clustering at the same time. However, time-lapse morphology alterations happened more slowly than in situations of step-broken Cu clustering in the 1 mbar CO(g) environment. This means that the kinetics of dissociated CO or atomic oxygen interactions was probably correlated with adsorbate-driven morphology changes under the reaction condition. Even though the direct comparison of kinetics measurements is not easy under the different reactor-like instruments between AP-XPS (very slow gas flow condition via a cone aperture) and AP-STM (the closed system by O-ring seals), the reviewer's request for additional labeling of the lapsed time is fair, to improve the readability of our manuscript for a broad readership in *Nature Communications*. We have revised the caption of Supplementary Fig. 16, as follows.

Supplementary Fig. 16:

Comparison observation results of CO₂ dissociation over vicinal Cu surfaces probed with AP-XPS and AP-STM. Collected Cu 2p, O 1s, and C 1s core-level AP-XP spectra in UHV (bottom; black line), 0.8 mbar CO₂(g) (middle; red line), and gas evacuation (top; blue line) conditions at 300 K ($h\nu = 1486.7$ eV). The Cu(997) surface was exposed to 0.8 mbar CO₂(g) in the analysis chamber for 214 minutes before recording the plotted AP-XPS spectra (middle; red line). (inset) Corresponding morphology structures of the Cu(997) surface in UHV, 1 mbar CO₂(g) [$t = t_0 + 77$ min], and gas evacuation conditions. The displayed *in situ* AP-STM images were sequentially obtained in the same manner as Figs. 1c-f at 300 K.

REVIEWERS' COMMENTS

Reviewer #3 (Remarks to the Author):

The authors successfully addressed the points I raised in the previous round. Therefore, I am happy to recommend its publication in Nature Communications.

Below is my comment on the authors' response to my comment #3-2. The authors' response is optional. The difference in O 1s and C 1s AP-XPS spectra of Cu(997) in 0.8 mbar CO₂ between this study and the previous work [Ref. 9 in the main text; T. Koitaya et al., Top. Catal. 59, 526-531 (2016)] was ascribed to the different probing depths (photon energies) in the XPS experiments. However, this claim requires further experimental confirmation by showing the probing depth dependence using SR X-rays. If the "surface" adsorbates (atomic O, CO₃, and CO) show the probing depth dependence, it suggests the presence of the sub-surface chemical species. The differences in experimental conditions such as flow rate, sample temperature, lapse-time, quality of background vacuum are other possible reasons for the difference in the AP-XPS spectra of CO₂/Cu(997). In the revised main text (Page 20, Line 5), it might be appropriate to change "This originates from the technical issue..." to "This could originate from the technical issue..."

Response to Reviewer #3:

General comment: *The authors successfully addressed the points I raised in the previous round. Therefore, I am happy to recommend its publication in Nature Communications.*

Below is my comment on the authors' response to my comment #3-2. The authors' response is optional. The difference in O 1s and C 1s AP-XPS spectra of Cu(997) in 0.8 mbar CO₂ between this study and the previous work [Ref. 9 in the main text; T. Koitaya et al., Top. Catal. 59, 526-531 (2016)] was ascribed to the different probing depths (photon energies) in the XPS experiments. However, this claim requires further experimental confirmation by showing the probing depth dependence using SR X-rays. If the "surface" adsorbates (atomic O, CO₃, and CO) show the probing depth dependence, it suggests the presence of the sub-surface chemical species. The differences in experimental conditions such as flow rate, sample temperature, lapse-time, quality of background vacuum are other possible reasons for the difference in the AP-XPS spectra of CO₂/Cu(997). In the revised main text (Page 20, Line 5), it might be appropriate to change "This originates from the technical issue..." to "This could originate from the technical issue..."

Response: We appreciate Reviewer 3 for the recommendation on our manuscript for publication in *Nature Communications*. As pointed out by the reviewer, we agree that various technical issues regarding the different experimental conditions would involve the identification of surface or sub-surface adsorbate species during AP-XPS measurements. We have revised the indicated sentence to address the reviewer's request very carefully, as below.

Revised sentence:

This could originate from the technical issue for characterizing CO₃* species, due to the different probing depths between the synchrotron-based photon source ($h\nu = 630$ eV) and monochromatic Al $K\alpha$ X-ray source ($h\nu = 1486.7$ eV), as well as other experimental conditions such as gas flow rate, sample temperature, lapsed-time, and background pressure in the analysis chamber.